# Asymmetric cell division shapes naive and virtual memory T-cell immunity during ageing

Mariana Borsa [1,2], Niculò Barandun[1,6], Fabienne Gräbnitz[1,6], Isabel Barnstorf[1], Nicolas S. Baumann[1], Katharina Pallmer[1], Samira Baumann [1], Dominique Stark [1], Miroslav Balaz[3], Nathalie Oetiker[1], Franziska Wagen[1], Christian Wolfrum [3], Anna Katharina Simon[2], Nicole Joller [4], Yves Barral [5], Roman Spörri [1] & Annette Oxenius [1✉]

Efficient immune responses rely on heterogeneity, which in CD8[+] T cells, amongst other mechanisms, is achieved by asymmetric cell division (ACD). Here we find that ageing, known to negatively impact immune responses, impairs ACD in murine CD8[+] T cells, and that this phenotype can be rescued by transient mTOR inhibition. Increased ACD rates in mitotic cells from aged mice restore the expansion and memory potential of their cellular progenies. Further characterization of the composition of CD8[+] T cells reveals that virtual memory cells (T$_{VM}$ cells), which accumulate during ageing, have a unique proliferation and metabolic profile, and retain their ability to divide asymmetrically, which correlates with increased memory potential. The opposite is observed for naive CD8[+] T cells from aged mice. Our data provide evidence on how ACD modulation contributes to long-term survival and function of T cells during ageing, offering new insights into how the immune system adapts to ageing.

[1] Institute of Microbiology, ETH Zürich, Zurich, Switzerland. [2] Medical Sciences Division, NDORMS, Kennedy Institute of Rheumatology, University of Oxford, Oxford, UK. [3] Institute of Food Nutrition and Health, ETH Zürich, Schwerzenbach, Switzerland. [4] Institute of Experimental Immunology, University of Zurich, Zurich, Switzerland. [5] Institute of Biochemistry, ETH Zürich, Zurich, Switzerland. [6] These authors contributed equally: Niculò Barandun, Fabienne Gräbnitz. ✉email: oxenius@microbio.ethz.ch

Efficient T-cell responses rely on heterogeneity. In the context of CD8[+] T cells, this is achieved both by a diverse naive T-cell repertoire and by the ability of single cells to generate effector and memory progenies, responsible for both clearance of intracellular pathogens and long-term protection against reinfections. Asymmetric cell division (ACD) is an important mechanism to generate T-cell diversity, and its loss contributes to impaired immune responses, particularly with regard to the formation and maintenance of a memory T-cell pool[1–4]. Upon ACD, a single T cell can give rise to daughter cells inheriting distinct contents of fate determinants through the polarized segregation of cell cargoes, which culminates in different potential fates. We reported previously a correlation between stemness and the ability of a CD8[+] T cell to undergo ACD by showing that short-lived effector and exhausted CD8[+] T cells divide mainly symmetrically, while naive and central memory pluripotent CD8[+] T cells are capable to divide asymmetrically[3].

The ageing of the immune system, a multifaceted phenomenon also known as immunosenescence, threatens T cell diversity, as a result of thymic involution and antigen exposure history. This culminates in both inefficient immune responses and increased susceptibility to autoimmunity[5–7]. In other cell types, such as haematopoietic stem cells (HSCs), the loss of diversity observed during ageing has been linked to their impaired ability to undergo ACD. In this scenario, the loss of asymmetric fates in HSCs is clearly detrimental for haematopoiesis and maintenance of stemness[8].

The decline of thymic functions observed in aged individuals results in a substantial change in T cell composition as fewer naive T cells emigrate to the periphery, and memory-phenotype CD8[+] T cells start to play a more dominant role as they accumulate due to cytokine-driven homoeostatic proliferation[9–11]. Not being previously exposed to foreign antigen, memory-phenotype CD8[+] T cells are usually termed virtual memory T cells (T$_{VM}$ cells)[12–19]. How T$_{VM}$ cells are generated has been a long-standing question, especially concerning whether this is a TCR-independent or TCR-instructed process. While T$_{VM}$ cells can be found in TCR-transgenic Rag deficient hosts[20] and in both pathogen-free and germ-free mice[21], a recent report identified a group of thymic precursors with self-reactive TCRs that give rise to memory-phenotype CD8[+] T cells[22]. In contrast to cognate antigen-experienced memory T cells, T$_{VM}$ cells show consistent low expression of the α4 integrin CD49d. Therefore, CD49d can be used as a suitable marker to discriminate between true and virtual memory lymphocytes[15,23,24].

T$_{VM}$ cells show paradoxical characteristics: their increased expression of CD122, a feature of memory cells, endows them with a competitive advantage in sensing IL-15, hence facilitating their long-term survival[14,25], but they are also efficient producers of IFNγ, a feature of effector cells, resulting in their major protective role against infections in aged individuals[26] despite their lower proliferative potential[27]. Indeed, T$_{VM}$ cells can readily produce cytokines in the absence of antigenic exposure upon stimulation with IL-12, IL-15 and IL-18[13,14,19,27], and constitutively express T-bet (a transcription factor that promotes the expression of effector genes) as well as effector molecules such as granzyme B and NKG2D (usually found in NK cells and activated T lymphocytes), features that facilitate their effector and bystander function[28].

As ageing drives a profound change in the composition of T cells, particularly due to the drop of naive T cells with a concommitant rise of T$_{VM}$ cells, it remains to be addressed whether the ability of CD8[+] T cells to undergo ACD is impacted by ageing. In this study, we found that ageing leads to an overall decline in the ability of CD8[+] T cells to undergo ACD. This phenomenon was due to the lower ACD rates found in naive CD8[+] T cells from aged animals, as CD8[+] T$_{VM}$ cells had this feature unaltered by ageing. As opposed to truly naive CD8[+] T cells from aged mice, T$_{VM}$ cells were unresponsive to mTOR inhibition-mediated enforcement of ACD, a strategy shown to promote ACD and memory potential in naive and memory CD8[+] T cells from young mice[3]. Functionally, CD8[+] T$_{VM}$ showed better re-expansion potential in adoptive transfer experiments compared to naive CD8[+] T cells from aged mice, providing new evidence of a correlation between the ability to divide asymmetrically and memory potential. Our data suggest that ACD might play an important role to counteract immunosenescence, in that it is retained in T$_{VM}$ cells but lost in aged naive cells, thereby preserving the ability to adopt differential fates in T$_{VM}$ cells.

## Results

**CD8[+] T cells from aged mice are less able to undergo ACD.** Considering the reduced capability of naive T cells to form memory populations during immunoscenescence, we investigated whether the ability of CD8[+] T cells to undergo ACD is affected by ageing. To this end, we used bulk CD8[+] T cells isolated from naive young (8–16 weeks old), middle-aged (>40 weeks old) and old (>70 weeks old) P14 mice (carrying a transgenic T-cell receptor specific for the lymphocytic choriomeningitis virus (LCMV) glycoprotein residues 33-41 (GP33) presented on H-2D[b]). The CD8[+] T cells were stimulated on plates coated with human Fc-ICAM-1, α-CD3 and α-CD28 for 36 h, allowing them to enter the first round of cell division, identifiable as different stages of mitosis by confocal microscopy. CD8 polarization on mitotic cells was used as a reliable readout of ACD[4]. Comparison of the ACD rates from naive CD8[+] T cells isolated from young or middle-aged/old P14 mice revealed that ageing correlated with an impaired ability to divide asymmetrically. We have shown previously that transient inhibition of the mTOR pathway by rapamycin is an efficient strategy to increase or restore the ability of CD8[+] T cells to divide asymmetrically[3]. Interestingly, ACD rates, comparable to young CD8[+] T cells, were restored when middle-aged/aged CD8[+] T cells were submitted to transient mTOR inhibition (Fig. 1). We observed a similar trend when measuring the polarized inheritance of both T-bet and phosphorylated S6[29–31] (Supplementary Fig. 1). Together, our data provide evidence that ageing impairs the ability of CD8[+] T cells to undergo ACD, and that transient mTOR inhibition can restore ACD in cells from aged mice. As ACD is correlated with stemness and memory potential[3,4,30,31], these results reveal a possible strategy to reinvigorate CD8[+] memory T-cell responses in the elderly.

**Re-establishment of ACD restores memory potential.** An increased ability to divide asymmetrically has been previously reported to promote memory potential of emerging CD8[+] T cell progeny[3]. Thus, we decided to test whether this would also hold true for cells from aged mice in which transient mTOR inhibition increased ACD rates in vitro. CD8[+] T cells isolated from aged P14 mice (>70 weeks) were stimulated under ACD conditions (+ICAM) in presence or absence of transient rapamycin treatment, and their progenies were adoptively transferred into naive and young wild-type recipient mice (Fig. 2A). Recipients were then acutely infected with 200 ffu LCMV-WE > 30 days after adoptive transfer, favoring maintenance of long-lived cells that can survive in absence of antigen, a feature of memory cells. We monitored the expansion of the adoptively transferred P14 cells in the blood, spleen and lymph nodes (LNs) of recipient mice (Fig. 2B, C, F). We observed higher frequencies and numbers of progenies derived from stimulation conditions in which mTOR was transiently inhibited with rapamycin. Phenotypic changes were only observed in P14 cells in the LNs, where transient rapamycin treatment during in vitro stimulation gave rise to

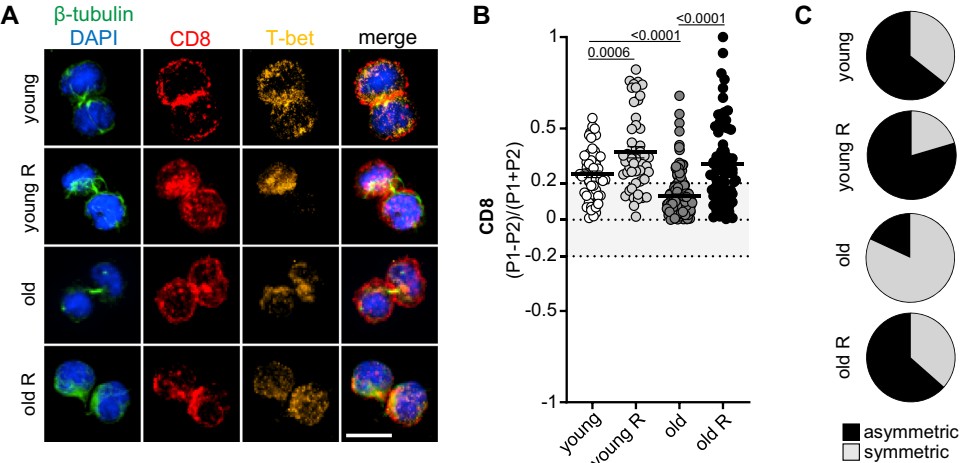

**Fig. 1 CD8+ T cells from aged mice exhibit decreased ability to undergo asymmetric cell division. A** Confocal images of murine CD8+ T cells fixed 36–40 h after in vitro stimulation on α-CD3, α-CD28 and human Fc-ICAM-1 coated wells (young, $n = 59$; young R, $n = 49$; middle-aged/old, $n = 171$; middle-aged/old R, $n = 74$). Transient mTOR inhibition was achieved by treatment with rapamycin (R) (20 nM). Cells were exposed to drug treatment from 12 h post activation until fixation for confocal microscopy analysis. Scale bar = 10 μm. **B** ACD rates observed in CD8+ T cells from middle-aged/aged and young mice upon transient mTOR inhibition with rapamycin. Data are represented as mean ± SEM. **C** Pie charts indicating frequencies of asymmetrically dividing cells from **B**. Level of CD8 asymmetric inheritance is depicted in black and level of CD8 symmetric inheritance in grey. Pooled data from three independent experiments. Age of the animals: experiments 1 and 2 (40–43 weeks) and experiment 3 (71 weeks). Statistical analysis was performed using the unpaired two-tailed Student's t test. Exact P values are depicted in the figure (see also Supplementary Fig. 1).

progenies with higher frequencies and numbers of cells committed to a memory fate, defined as KLRG1lo IL-7Rαhi expressing cells (Fig. 2D, G). Furthermore, progenies originating from higher asymmetry rate conditions, imposed by transient mTOR inhibition, exhibited increased cytokine production (TNF; Fig. 2E). As total P14 cell numbers were higher in progenies of cells transiently treated with rapamycin, we could also observe a significantly higher number of both IFNγ and TNF producing cells in this group. Together, these data provide further evidence regarding a correlation between increased ACD rates and progenies with increased memory potential, as determined by their enhanced survival and recall response upon re-challenge.

**T_VM cells have a unique proliferative and metabolic phenotype**. Our analysis of CD8+ T cells isolated from young and old P14 mice revealed an impaired ability of CD8+ T cells from aged mice to undergo ACD. In view of the heterogeneity of CD8+ T cells present in "naive" mice, we decided to assess phenotypical differences between cells derived from young or old mice. To that end, we analyzed the expression of CD44 in CD8+ T cells from the blood of young (<16 weeks old) and aged (>70 weeks) P14 mice, as previous work has shown that ageing leads to an increased CD44hi/CD44lo T-cell ratio[32–34]. The frequencies of CD44hi cells were significantly higher within the CD8+ T cell pool in aged mice (Fig. 3A). A previous study provided evidence of a correlation between a higher expression of CD44 and an exhausted signature during ageing[32]. Considering the lower asymmetry rates observed in CD8+ T cells from aged mice and the negative effect of T cell exhaustion on the ability of CD8+ T cells to divide asymmetrically[3], we characterized whether CD44hi cells would exhibit an exhausted phenotype. We stained peripheral blood CD8+ T cells from young and old P14 mice for the expression of both the co-inhibitory molecule PD-1[35–37], and the exhaustion marker CD39[38]. In agreement with more recent studies[27], despite a trend towards an increase in PD-1hi and CD39+ cells amongst CD44hi CD8+ T cells in aged animals, these cells were still a minority in the CD44hi compartment (Supplementary Fig. 2). These data exclude exhaustion as a direct or major cause for the lower ACD rates observed in CD8+ T cells from aged mice.

CD44hi CD8+ T cells are known to accumulate with ageing[39], and several reports have identified a large proportion of these cells as virtual memory cells (T_VM)[14,18,19,26,27]. To investigate whether the CD44hi cells that were found at higher frequencies in aged P14 mice were indeed T_VM cells, we first performed a phenotypic characterization[16]. We collected blood from both young and old unimmunized P14 mice and analyzed the expression of CD44, CD122, CXCR3, NKG2D and CD49d in CD8+ T cells. Confirming the T_VM cell phenotype, CD44 expression on CD8+ T cells was accompanied by high expression of CD122, CXCR3 and NKG2D, and low expression of CD49d (Fig. 3A). Furthermore, as the CD44hi CD8+ T cells were not CD49dhi in either young or aged unimmunized P14 mice, we excluded them being antigen-driven memory cells. Other studies have shown an accumulation of true memory CD8+ T cells upon ageing in wild-type animals with a polyclonal lymphocyte repertoire (Quinn et al.[27]). The absence of this population in naive P14 mice, independent of their age, corroborated that T_VM cells derive from naive cells in a mechanism distinct from the differentiation of "true" memory cells by cognate antigen recognition.

To further characterize the T_VM cell population present in the blood of aged P14 animals, we separately analyzed the proliferation profiles of naive and T_VM splenic CD8+ T cells isolated from young and aged mice 42 or 72 h post activation (Fig. 3B). At an early time point (42 h post activation), T_VM cells from both young and aged animals had undergone fewer divisions in comparison to their naive counterparts. However, 72 h post activation, T_VM cells isolated from young hosts exhibited the highest proliferative capacity among all tested cell types, while T_VM cells from aged hosts were the ones showing the lowest proliferation speed, but still similar to the profile seen in naive CD8+ T cells from young or aged animals. Truly naive CD8+ T cells from young and aged animals showed very similar proliferation profiles at both timepoints analyzed. These data corroborate previous observations, showing that T_VM cells from young hosts exhibit faster cell cycling in comparison to naive

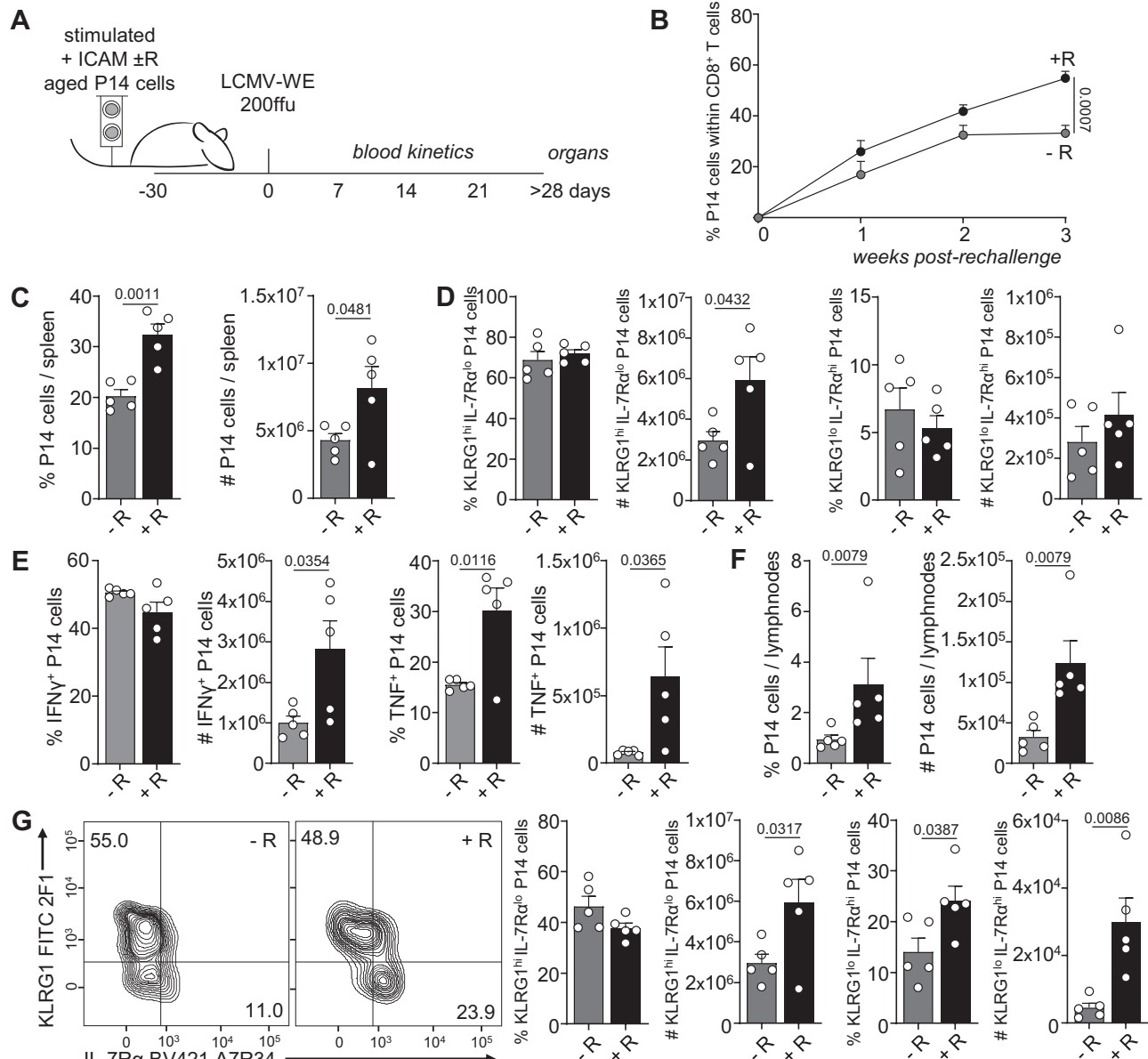

**Fig. 2 Re-establishment of asymmetric cell division in CD8+ T cells from aged mice correlates with increased memory potential. A** Experimental setup (illustration by MB). Naive P14 cells from aged mice were stimulated in vitro on α-CD3, α-CD28 and human Fc-ICAM-1 coated wells in presence or absence of transient rapamycin (R) treatment, harvested after 36 h, and adoptively transferred into young and naive wild-type recipients. Each recipient mouse received $1 \times 10^4$ cells and was infected intravenously with LCMV-WE > 30 days later. **B** Frequencies of P14 cells within the CD8+ T-cell population in the blood. **C** Frequency and numbers of adoptively transferred P14 cells within CD8+ T cells in the spleens of recipient mice. **D** Frequencies and numbers of KLRG1hi IL-7Rαlo or KLRG1lo IL-7Rαhi P14 cells in the spleens of recipient mice. **E** Frequencies and numbers of splenic IFNγ and TNF P14 producing cells. **F** Frequencies and numbers of adoptively transferred P14 cells within CD8+ T cells in the inguinal lymph nodes of recipient mice. **G** Representative flow cytometry plots of KLRG1 and IL-7Rα expression in P14 cells in the lymph nodes of recipient mice 30 days after LCMV-WE challenge (left panel); percentages and numbers of KLRG1hi IL-7Rαlo and KLRG1lo IL-7Rαhi P14 cells in the lymph nodes of recipient mice (right panel). Representative data (n = 5 per group) from one out of two experiments. Aged P14 animals used for adoptive cell transfer: 95 weeks (experiment 1); 71 weeks (experiment 2). Data are depicted as mean + SEM. Statistical analysis was performed using the unpaired two-tailed Student's t test (**B–E**, **G**, week 3) or when data did not pass the normality test, the unpaired two-tailed Mann–Whitney U test (**F**, **G**, #KLRG1hi IL-7Rαlo). Exact P values are depicted in the figure.

cells[27], but also suggest a unique cell division kinetics of $T_{VM}$ cells, as the time for the first division seemed to be extended, especially in the ones originating from an ageing host. Cell cycling speed can be a result of distinct metabolic activities and plays a role in T cell differentiation[40–42]. It has been reported that resting $T_{VM}$ CD8+ T cells have a different metabolism in comparison to naive or true memory cells, exhibiting elevated

spare respiratory capacity (SRC)[25]. We therefore investigated whether activated naive and $T_{VM}$ CD8+ T cells exhibited metabolic differences. Because frequencies of $T_{VM}$ cells were particularly rare in young P14 mice, these cells were not used for metabolic assays. We assessed the metabolic performance of naive CD8+ T cells from young and aged animals, and $T_{VM}$ cells from aged animals 36 h post activation (when cells either did not divide

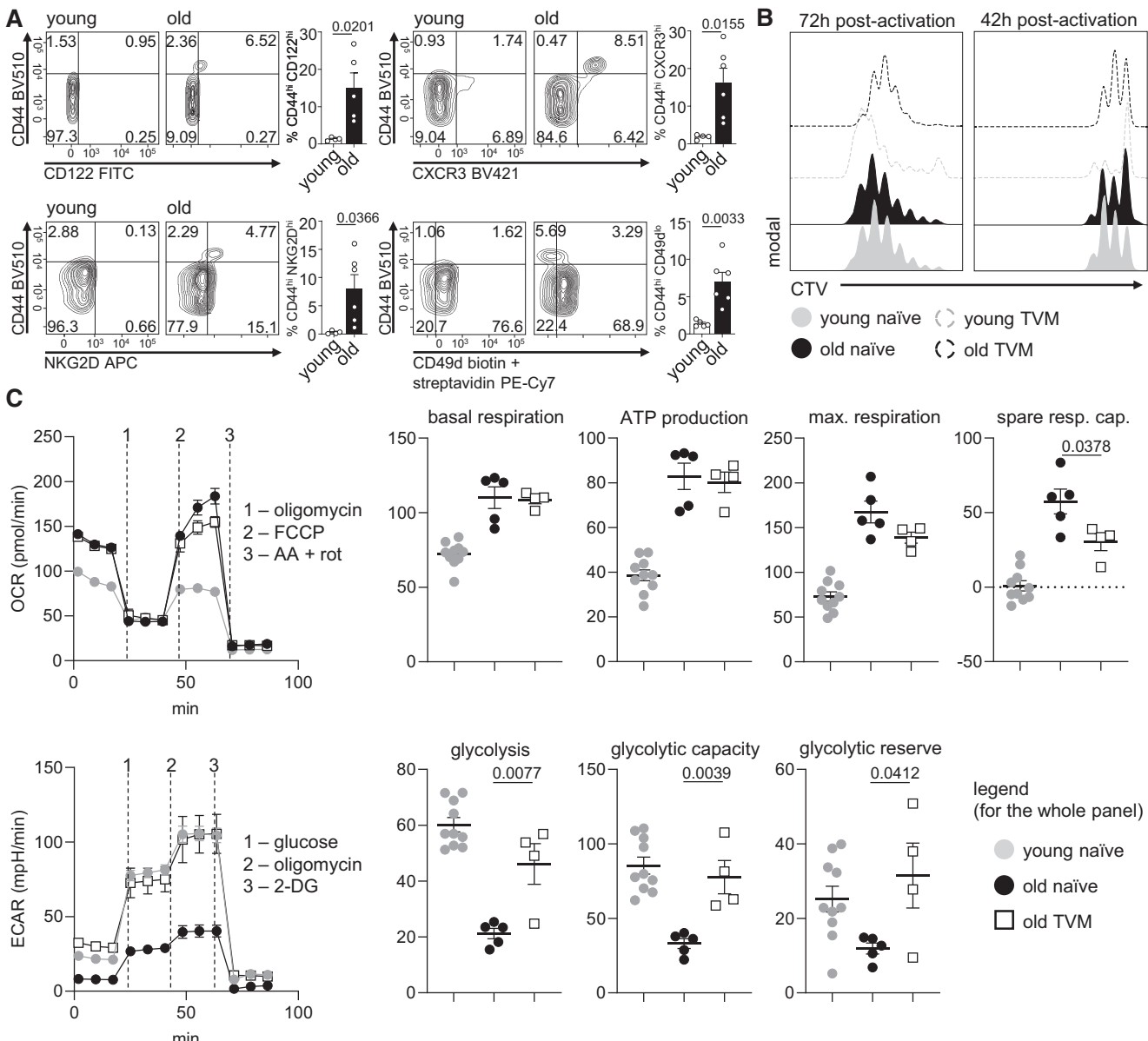

**Fig. 3 T$_{VM}$ cells have a unique proliferative and metabolic phenotype. A** Representative flow cytometry plots showing CD122, CXCR3, NKG2D and CD49d expression on CD8$^+$ T cells in the blood of young and old P14 mice. Frequencies of CD44$^{lo}$ and CD44$^{hi}$ CD8$^+$ T cells expressing high CD122, CXCR3 and NKG2D levels and low CD49d levels are depicted in the bar graphs. Pooled data from two independent experiments (young, $n = 4$: old, $n = 5$; aged mice: 70, 90 or 100 weeks old). **B** Histograms depicting proliferation of purified naive or T$_{VM}$ CD8$^+$ T cells from young or aged mice (CTV-labelled, stimulated on α-CD3, α-CD28 and human Fc-ICAM-1 coated wells). After 42 h, the proliferation profile was analyzed by CTV dilution (left panel) or cells were transferred to uncoated wells for further 30 h in medium containing IL-2, IL-7 and IL-15, until their proliferation kinetics was assessed 72 h post stimulation (right panel). Representative data from one out of two experiments (aged mice: 70 or 82 weeks old). **C** Oxygen consumption rate (OCR) (upper panel) and extracellular acidification rate (ECAR) (lower panel) of activated CD8$^+$ T cells was measured under basal conditions and in response to indicated drugs. Representative data from one out of two experiments, where data points are technical replicates (young naive, $n = 10$; old naive, $n = 5$, old TVM, $n = 4$; aged mice >100 weeks old in all experiments). Data are depicted as mean ± SEM. Statistical analysis was performed using the unpaired two-tailed Student's $t$ test. Exact $P$ values are depicted in the figure.

or underwent their first round of mitosis) using Seahorse assays. Both naive and T$_{VM}$ CD8$^+$ T cells from aged animals outperformed naive cells from young individuals concerning their SRC (Fig. 3C, upper panel). T$_{VM}$ cells from aged individuals showed a similar glycolytic performance as young naive cells, both with respect to their glycolytic capacity and glycolytic reserve, features for which naive cells from aged hosts exhibited impaired function (Fig. 3C, lower panel). This could indicate that activated T$_{VM}$ cells from aged hosts are metabolically adapted to swiftly respond to environmental changes through their ability to

use both glycolysis and oxidative phosphorylation (OXPHOS) in order to sustain their metabolic demands, while simultaneously being "slow-dividers", a feature previously reported to benefit ACD[3] and memory formation[42].

**T$_{VM}$ CD8$^+$ T cells show intrinsically high ACD rates**. The observation of phenotypic, functional and metabolic differences between naive and T$_{VM}$ CD8$^+$ T cells led us to investigate whether there would also be differences in their ability to divide

asymmetrically. To that end, we sorted CD8$^+$ T cells isolated from spleens and LNs of young and aged P14 mice into naive (CD44$^{lo}$) and T$_{VM}$ (CD44$^{hi}$, CD49d$^{lo}$) populations. Cells were then stimulated in vitro under ACD conditions and submitted to transient mTOR inhibition with rapamycin or Akt-kinase inhibitor, known to increase ACD rates in naive and memory CD8$^+$ T cells from young individuals[3]. The separation into naive and T$_{VM}$ cells prior to stimulation allowed us to directly compare their abilities to undergo ACD and to discriminate which cells were responsive to mTOR inhibition, evoking higher ACD rates and exhibiting improved memory potential after adoptive transfer (Figs. 1 and 2). T$_{VM}$ CD8$^+$ T cells from both young and aged mice showed intrinsically high ACD rates, based on the levels of CD8 polarization (Fig. 4B, C). By contrast, naive CD8$^+$ T cells from old mice had a markedly impaired capacity to undergo ACD in comparison to naive CD8$^+$ T cells from young mice (Fig. 4A, C). As T$_{VM}$ cells take longer to undergo their first mitosis, it is likely that the frequencies of dividing T$_{VM}$ cells analysed amongst the bulk CD8$^+$ T-cell population at 36 h post stimulation (Fig. 1) are under-represented. Furthermore, we cannot exclude that the analysed mitotic cells might represent a pool of T$_{VM}$ cells that retains their proliferation potential. These results, however, suggest that naive CD8$^+$ T cells are responsible for the overall lower ACD rates observed in total CD8$^+$ T cells isolated from aged P14 mice in comparison to their young counterparts (Fig. 1). The intrinsically high ACD rates in T$_{VM}$ CD8$^+$ T cells could not be further increased by both rapamycin and Akt-kinase inhibitor treatment, while CD44$^{lo}$ naive CD8$^+$ T cells from both young and old mice showed increased ACD rates under the same conditions (Fig. 4C). The responsiveness of naive CD8$^+$ T cells from old mice to rapamycin treatment explains the re-established asymmetry observed when total CD8$^+$ T cells from old P14 mice were exposed to rapamycin treatment (Fig. 1). We speculate that the failure in modulating ACD in T$_{VM}$ CD8$^+$ T cells might result from both their higher mTOR activity in comparison to the their naive counterparts, and the inefficacy of the inhibitors, at least in the concentrations that they were administered, in down-regulating this activity, especially what concerns downstream targets of mTORC2 (Supplementary Fig. 3). As ACD relies on the formation of a stable immune synapse, we analyzed the expression of CD3, CD28 and LFA-1 in naive and T$_{VM}$ CD8$^+$ T cells, and observed that the latter exhibited significantly higher levels of LFA-1 on the surface, perhaps contributing to their intrinsically high ACD rates (Supplementary Fig. 4). We obtained similar results when performing experiments with polyclonal CD8$^+$ T cells from wild-type C57BL/6 mice (Supplementary Fig. 5). Considering the superior memory potential exhibited by progenies of cells able to divide more asymmetrically, these results suggest that impaired immune responses in the elderly can be partially rejuvenated by re-establishing their ability to undergo ACD in naive CD8$^+$ T cells through transient mTOR inhibition. In addition, as ACD proved to be beneficial for establishment of immune memory, the intrinsic ability of T$_{VM}$ CD8$^+$ T cells to divide asymmetrically might be a compensatory mechanism to sustain efficient T-cell responses upon ageing.

**ACD modulation targets naive CD8$^+$ T cells from aged mice.** Within CD8$^+$ T cells from old P14 mice, the naive subset showed strongly impaired ability to undergo ACD, but simultaneously the best responsiveness to transient mTOR inhibition as a strategy to re-establish ACD. To investigate a potential improvement of immune responses by modulating ACD in these naive CD8$^+$ T cells, we adoptively transferred progenies of stimulated naive and T$_{VM}$ P14 cells from young and old mice into naive wild-type recipients (Fig. 5A). Stimulation was performed

under ACD conditions, with or without transient mTOR inhibition with rapamycin. After 30 days, recipients were infected with LCMV-WE and numbers of P14 cells were evaluated in the spleen 30 days post viral challenge (Fig. 5B, C). Progenies of rapamycin treated naive P14 cells from both young and old P14 mice were present in higher numbers than their untreated counterparts (Fig. 5B). Progenies of CD8$^+$ T$_{VM}$ cells from both young and aged mice expanded to comparable frequencies or total numbers in presence or absence of mTOR inhibition during in vitro stimulation. As mTOR inhibition could not enhance their ability to undergo ACD, these data strengthen the correlation between the ability to undergo ACD and the potential to form a pool of memory cells. Rapamycin treatment did not lead to significant changes in the phenotypes of the resulting progenies, however, at least for naive cells from young mice, there was a trend towards higher frequencies of KLRG1$^{lo}$ IL-7Rα$^{hi}$ and lower frequencies of KLRG1$^{hi}$ IL-7Rα$^{lo}$ cells of the rapamycin treated group (Fig. 5C), as previously reported by us[3]. Interestingly, progenies of aged CD8$^+$ T$_{VM}$ cells exhibited significantly lower frequencies of KLRG1$^{lo}$ IL-7Rα$^{hi}$ cells, with concomitant emergence of a double-positive population for KLRG1 and IL-7Rα, known to be effector-like memory cells that exhibit enhanced cytotoxic activity[43] (Fig. 5C). Functionally, frequencies of IFNγ and TNF producing cells were not affected by cell type or ageing (Fig. 5D). As total P14 cell numbers were higher in progenies of naive CD8$^+$ T cells transiently treated with rapamycin and in progenies of T$_{VM}$ cells (independent of previous rapamycin treatment), we also observed significantly elevated numbers of both IFNγ and TNF producing cells in these groups. Together, these results emphasize a link between ACD, memory potential and efficient immune responses, and provide new insights into how the ability to undergo ACD is affected by ageing. We propose that CD8$^+$ T$_{VM}$ cells might represent an adaptation to immunosenescence and compensate for the loss of ACD potential in naive cells. Furthermore, as the memory potential of naive CD8$^+$ T cells from aged hosts could be increased by transient mTOR inhibition and increased ACD rates, these data reinforce the potential of this intervention for translational purposes in regenerative medicine.

**Discussion**

Ageing is associated with a progressive decline in the efficiency of mounting effective and long-lasting immune responses and results in higher vulnerability to infections in both mice and humans[44]. Virtually all aspects of immunity are affected by ageing, from the initial contact with a pathogen to its clearance or coexistence with a persisting pathogen[7]. Concerning T cell responses, the most profound change observable upon ageing is thymic involution[10,45], which leads to marked changes in peripheral T cell maintenance mechanisms. With the decrease of thymic output, T cells that are already in the periphery go through more rounds of homoeostatic proliferation. Amongst possible other factors yet to be discovered, this contributes to the naive-to-memory conversion of their phenotype, which is independent of cognate antigen encounter, hence these cells are referred to as virtual memory cells[13–15,17–19,27]. T$_{VM}$ CD8$^+$ T cells accumulate with ageing and depend on homoeostatic cytokines for maintenance. Besides activation through the T-cell receptor to exert effector functions, T$_{VM}$ CD8$^+$ T are also responsive to cytokine activation[14,19,28].

In the present study, our aim was to investigate a new aspect of the ageing immune system and characterize the ability of CD8$^+$ T cells to undergo ACD. Our results on bulk CD8$^+$ T cells show that total naive cells from aged mice divide less asymmetrically than their young counterparts. We have previously developed a

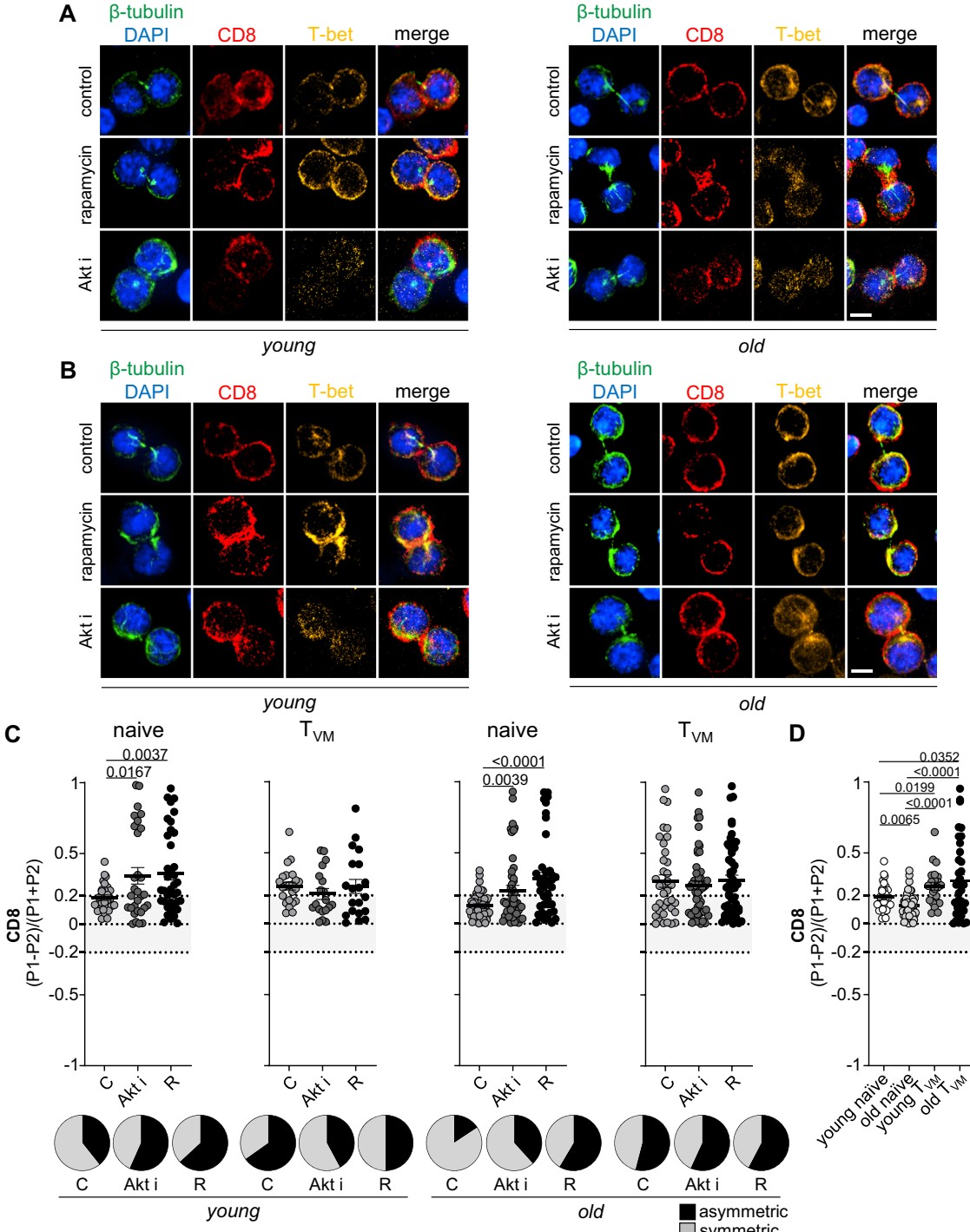

**Fig. 4 T_VM CD8+ T cells show intrinsically high ACD rates. A** Confocal images from murine naive CD8+ T cells. **B** Confocal images from murine T_VM CD8+ T cells. Cells were fixed 36–40 h after in vitro stimulation on α-CD3, α-CD28 and human Fc-ICAM-1 coated wells. Transient mTOR inhibition was achieved by treatment with rapamycin (20 nM) or Akt-kinase inhibitor (5 µM). Cells were exposed to drug treatment from 12 h post activation until fixation for confocal microscopy analysis. Scale bar = 5 µm. **C** CD8 asymmetry rates in naive and T_VM CD8+ T cells isolated from young or old mice (young naive C, n = 28; young naive Akt i, n = 30; young naive R, n = 38; young T_VM C, n = 23; young T_VM Akt i, n = 19; young T_VM R, n = 20; old naive C, n = 51; old naive Akt i, n = 47; old naive R, n = 42; old T_VM C, n = 37; old T_VM Akt i, n = 51; old T_VM R, n = 52). Frequencies of asymmetrically dividing cells are depicted in pie charts on the right side of each graph. **D** Comparison of CD8 polarization amongst young naive, old naive, young T_VM and old T_VM CD8+ T cells (young naive, n = 28; old naive, n = 51; young T_VM, n = 23; old T_VM, n = 37). Data are shown as mean ± SEM. Pooled data from five independent experiments (aged mice: 71, 95, 97, 102 or 108 weeks old). For each experiment sorted cells from at least two animals were pooled. Statistical analysis was performed using the unpaired two-tailed Student's *t* test. Exact *P* values are depicted in the figure.

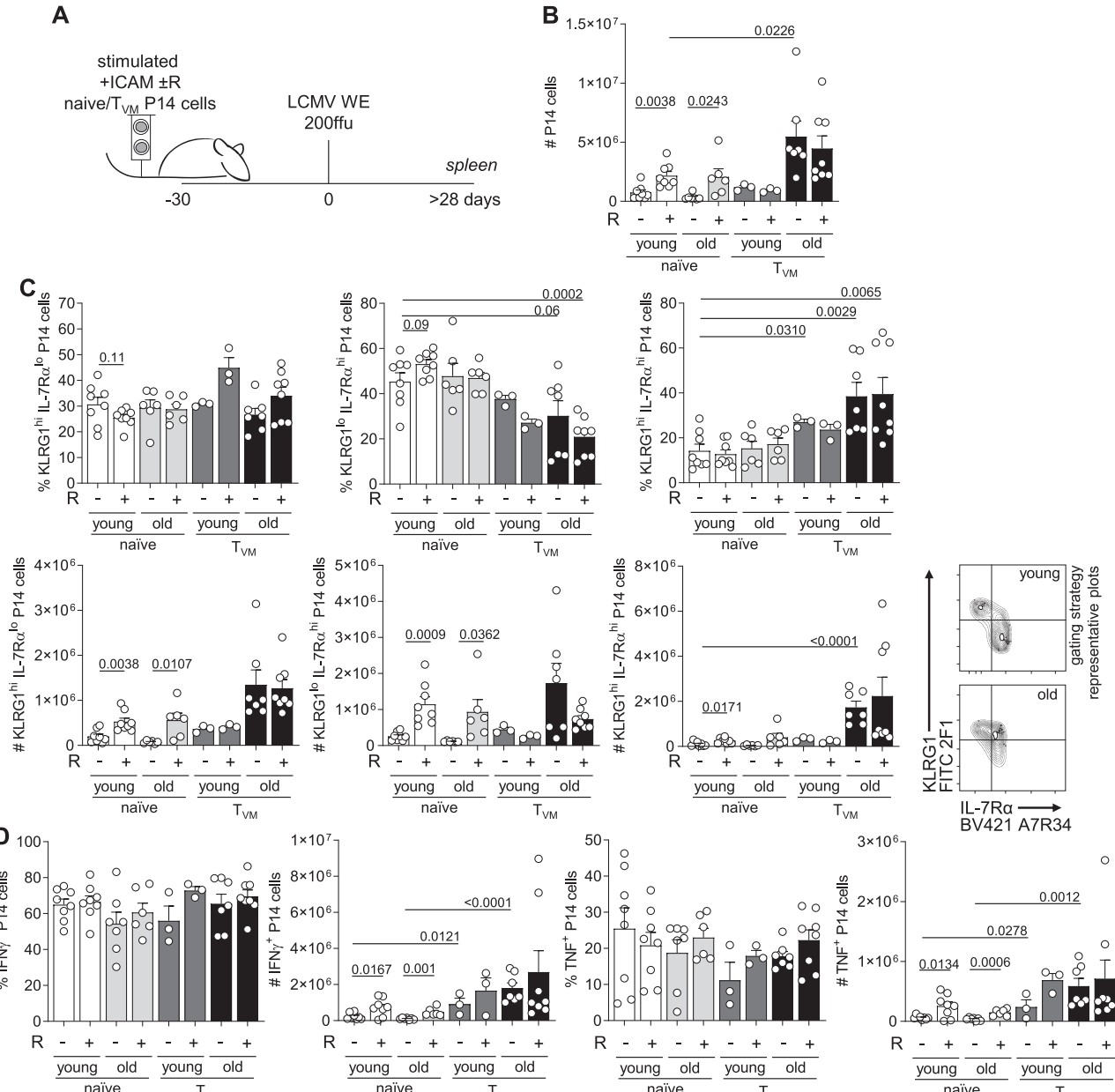

**Fig. 5 Re-establishment of ACD by transient mTOR inhibition in naive CD8$^+$ T cells from aged mice reflects in correlates with improved memory potential. A** Experimental setup (illustration by MB). Naive and $T_{VM}$ CD8$^+$ T cells were sorted from young and old P14 mice spleens, stimulated under different conditions, harvested after 36 h and adoptively transferred into wild-type recipients. Each recipient mouse received 1 × 10$^4$ cells and was infected intravenously with LCMV-WE (200 ffu) 30 days later. **B** Numbers of adoptively transferred P14 cells within CD8$^+$ T cells in the spleens of recipient mice. **C** Percentages and numbers of KLRG1$^{hi}$ IL-7Rα$^{lo}$, KLRG1$^{lo}$ IL-7Rα$^{hi}$ and KLRG1$^{hi}$ IL-7Rα$^{hi}$ P14 cells in the spleens of recipient mice (left panel). Representative plots showing gating strategy used to access expression of KLRG1 and IL-7Rα (right panel). **D** Frequencies and numbers of IFNγ, TNF and IL-2 producing cells in the spleens of recipient mice. Pooled data from two independent experiments (young naïve, $n = 8$; young naïve R, $n = 8$; young $T_{VM}$, $n = 3$; young $T_{VM}$ R, $n = 3$; old naïve, $n = 6$; old naïve R, $n = 6$; old $T_{VM}$, $n = 7$; old $T_{VM}$ R, $n = 8$). Aged P14 animals used for adoptive cell transfer: 92 or 96 weeks (experiment 1); 70 or 75 weeks (experiment 2). Data are depicted as mean + SEM. Statistical analysis was performed using the unpaired two-tailed Student's $t$ test. Exact $P$ values are depicted in the figure.

strategy to increase ACD rates by transient mTOR inhibition with either rapamycin or Akt-kinase inhibitor, which was based on the reported benefits of mTOR inhibition on memory formation[46,47]. This transient mTOR inhibition during in vitro stimulation proved to be efficient in enhancing memory potential of progenies of naive, memory and PD-1$^{int}$ Tcf1$^+$ exhausted CD8$^+$ T cells[3]. Of relevance, mTOR is a key regulator of ageing in different organisms, from yeast to mammals[48], and its inhibition

has been linked to extended life span and improved immune functions in old mice and humans[49,50]. Furthermore, the inhibition of the homologous pathway TOR in yeast contributes to rejuvenation of yeast cells through higher ACD rates[51]. Transient rapamycin treatment of CD8$^+$ T cells from aged mice indeed led to increased ACD rates, confirming our initial hypothesis that ACD can be modulated in these cells. This remarkable result proved to be of functional relevance, as progenies originating

from enforced ACD conditions exhibited improved memory potential in a viral challenge model.

In line with previous reports, we observed a marked increase in the CD44$^{hi}$/CD44$^{lo}$ ratio with ageing[32–34], and identified the emerging CD44$^{hi}$ population as T$_{VM}$ cells. Besides phenotypic differences with naive T cells in surface marker expression, T$_{VM}$ cells also exhibited different proliferation rates and metabolism. After stimulation in vitro, T$_{VM}$ cells from both young and aged animals required a longer time to undergo their first rounds of division in comparison to their naive counterparts. However, the opposite was observed at later timepoints, when progenies of T$_{VM}$ cells showed similar or higher proliferation rates. Activated T$_{VM}$ cells/progenies acquired a unique effector-memory mixed metabolic profile, being capable to rely on both glycolysis and OXPHOS.

The differences in observed phenotype, proliferation kinetics and metabolism between naive and T$_{VM}$ CD8$^+$ T cells led us to investigate which cell type is responsible for the lower ACD rates found in total CD8$^+$ T cells originating from aged mice. Surprisingly, the ability of CD44$^{lo}$ naive CD8$^+$ T cells to undergo ACD was strongly affected by ageing, as CD8$^+$ T cells from old mice exhibited low ACD rates compared to their young counterparts. This defect could be reversed by transient mTOR inhibition. As expected, increased ACD rates observed in vitro led to improved function and correlated with a higher re-expansion potential upon antigenic challenge in vivo. By contrast, T$_{VM}$ cells exhibited intrinsically high ACD rates, which could not be further increased by transient mTOR inhibition. As ACD is thought to be beneficial for diversified T-cell responses, in particular to impart memory potential, we propose that T$_{VM}$ cells are an adaptation of the immune system to immunesenescence. Indeed, our results might explain how T$_{VM}$ cells end up comprising the majority of the CD44$^{hi}$CD62L$^{hi}$ T$_{CM}$ cell population in LCMV infected mice[25], or how these cells are able to compose the majority of the central memory CD8$^+$ T cell (T$_{CM}$) compartment after LM-OVA antigenic challenge, while also being able to generate effector responses[19].

The better expansion potential of T$_{VM}$ CD8$^+$ T-cell progenies in comparison to their naive counterparts in adoptive transfer experiments followed by antigenic challenge contradicts previous findings, where CD44$^{lo}$ naive CD8$^+$ T cells were described as superior in that context[27,32]. However, the divergent results may be explained by the different experimental setups. In our study, adoptively transferred cells were progenies of T$_{VM}$ cells that underwent ACD and were required to survive for over 30 days before re-challenge. In previous studies[27,32], recipient mice received unstimulated cells and were either immediately immunized with cognate antigen or left untreated to assess the differential survival of naive CD8$^+$ T cells and T$_{VM}$ CD8$^+$ T cells.

Taken together, our results suggest that the extended time taken by T$_{VM}$ cells to undergo their first rounds of division are beneficial for ACD, and thus for the establishment of asymmetric fates, which potentially also explains their exclusive proliferation and mixed metabolic phenotypes. ACD contributes to the simultaneous rise of (1) a pool of memory cells reliant on OXPHOS and exhibiting elevated SRC, and (2) a pool of effector cells that relies on glycolysis, and undergoes rapid proliferation at later timepoints[42]. We observed that T$_{VM}$ cells from aged animals, which exhibit higher expression of CD122 than their naive counterparts, are better survivors under limiting IL-15 concentrations in vitro, highlighting a potential role of survival in the observed increased recovery of adoptively transferred cells upon re-challenge/re-expansion (Supplementary Fig. 6). Thus, the virtual memory pool of CD8$^+$ T cells might present an adaptation of the ageing immune system to maintain a memory-like pool of

cells in absence of previous antigen encounter without compromising effector responses.

A question that remains unresolved is the reason for the refractoriness of T$_{VM}$ CD8$^+$ T cells to mTOR inhibition when measured by modulation of ACD rates. CD44 engagement is known to lead to downstream phosphorylation of Akt, resulting in overall higher mTOR activity[52]. When we assessed mTORC1 and mTORC2 activities, measured by phosphorylation levels of S6 and NF$\kappa$B[53], our data indicated that T$_{VM}$ cells have higher basal mTOR activity in comparison to their naive CD44$^{lo}$ counterparts and are refractory to mTORC2 inhibition (Supplementary Fig. 3). As there is evidence on the role of mTORC2 inhibition as an efficient strategy for ACD modulation[3], these data suggest that constitutive mTORC2 activity might be the reason for both the lack of ACD modulation in T$_{VM}$ cells and their efficient glycolytic metabolism. Furthermore, the inability to modulate ACD in TVM CD8$^+$ T cells, by either rapamycin or Akt-kinase inhibitor treatment, could be caused by the altered role of mTOR during CD8$^+$ T cell senescence, which potentially leads to differential activation of its downstream targets[54–56]. However, the reason for their higher intrinsic ability to divide asymmetrically remains unresolved and needs to be further investigated.

Transient mTOR inhibition efficiently restored the ability of naive CD8$^+$ T cells from aged mice to undergo ACD. This correlated with an increased re-expansion of transferred cells upon LCMV challenge, which is evidence of their better survival capacity and memory potential. Re-establishment of ACD might therefore represent a strategy for rejuvenation of senescent naive CD8$^+$ T cells found in aged individuals. Conversely, T$_{VM}$ cells exhibited intrinsically high ACD rates which could not be further increased by transient mTOR inhibition. Functionally, these properties culminated in increased memory potential shown by adoptive transfer experiments, providing a further link between ACD and efficient immunity. The accumulation of T$_{VM}$ cells with ageing might therefore, at least to some extent, compensate for the diminished memory potential of naive cells. Reinforcing ACD in these naive cells in aged individuals might offer new perspectives for improvement of immune functions in the elderly, and brings new insights on how the immune system adapts to ageing.

## Methods

**Ethics statement**. Animal experiments were performed in accordance with institutional policies, Swiss and British federal regulations, and were approved by the veterinary office of the Canton of Zürich (animal experimental permissions: 147/2014 and 115/2017) or by a local ethical review committee in UK (project license PPL 30/3388).

**Mice**. P14 mice (Rag sufficient) expressing a transgenic T-cell receptor specific for the glycoprotein 33-41 (gp33) epitope of the LCMV presented on H-2D$^{b}$[57] on a congenic CD45.1 background were bred and maintained at the ETH Phenomics Center. CD45.2 C57BL/6 were bred at the ETH Phenomics Center or purchased from Janvier Elevage (Saint Berthevin, France) or Charles River (UK). Six-to-sixteen-week-old male or female mice were considered young, mice > 50 weeks of age were considered middle-aged and mice > 70 weeks were considered aged. For all experiments where adoptive cell transfer was performed, aged mice were >70 weeks old. All mice were bred and maintained under specific pathogen-free conditions at 24 °C and 50% humidity, exposed to a 12:12 h light/dark cycle and kept in individually ventilated cages with autoclaved water and irradiated pellet feed provided ad libitum.

**Virus, viral peptides and infections**. LCMV peptide gp33-41 (gp33; KAVYN-FATM) was purchased from EMC microcollections GmbH (Tübingen, Germany). LCMV-WE was originally provided by R.M. Zinkernagel (University Hospital, Zurich, Switzerland) and was propagated on BHK-21 cells[58]. Acute infections were performed by intravenously injecting mice into the tail vein with 200 ffu of LCMV-WE.

**CD8$^+$ T-cell isolation.** CD8$^+$ T cells were isolated from spleens of CD45.1 P14 mice using the EasySep™ Mouse CD8$^+$ T cell Isolation Kit (Stemcell, Grenoble, France) or the MojoSort™ Mouse CD8$^+$ T cell Isolation Kit (BioLegend, Lucerna Chem AG, Luzern, Switzerland), following manufacturer's instructions. For analysis of specific cell subsets and CD8$^+$ T-cell enrichment, erythrocytes were lysed and remaining splenocytes were incubated with α-B220 biotin (RA3-6B2, BioLegend, 1/200), and α-CD4 biotin (GK1.5, BioLegend, 1/200) antibodies for 20 min at room temperature, followed by incubation with MojoSort™ streptavidin magnetic beads (BioLegend) for 5 min at room temperature and further magnetic separation. After enrichment, cells were then stained for phenotypical markers, and subpopulations of interest were sorted on a FACS Aria cell sorter (BD Biosciences).

**CD8$^+$ T-cell in vitro stimulation and adoptive transfer.** For transfer of daughter cells, when confocal microscopy analysis of mitotic cells was performed or when cell proliferation was assessed, cells were cultured in T-cell medium (RPMI 1640 (Bioconcept), 2 mM L-Glutamine (Bioconcept), 2% penicillin-streptomycin (Sigma-Aldrich), 10% foetal bovine serum (Omnilab), 25 mM HEPES (Gibco, Life Technologies, Zug, Switzerland), 1× non-essential amino acids (Sigma-Aldrich), 50 μM β-Mercaptoethanol (Gibco), 1 mM sodium pyruvate (Gibco) supplemented with self-made human IL-2, and stimulated on plate-bound human Fc-ICAM-1 (50 μg/ml) (R&D Biosciences, Bio-Techne AG, Zug, Switzerland), α-CD3 (5 μg/ml) (145-2C11, BioLegend) and α-CD28 (5 μg/ml) (37.51, BioLegend) for 30–36 h. mTOR modulation was done by adding 20 nM of rapamycin (Santa Cruz, Lab-Force AG, Muttenz, Switzerland) or 1–2 μM of Akt-kinase inhibitor (Sigma-Aldrich) 12 h post stimulation. For adoptive transfer, 1 × 10$^4$ purified CD45.1 P14 cells (stimulated as indicated) were intravenously injected into naive C57BL/6 CD45.2 recipient mice. For cell proliferation assays, cells were stained with Cell-Trace Violet™, CellTrace Yellow™ or CellTrace Violet™ (Life Technologies) following manufacturer's instructions prior to stimulation.

**Stimulation of lymphocytes and flow cytometry.** Blood samples used for kinetics analysis were obtained from the tail vein. Spleens were obtained from PBS-perfused mice. Single-cell splenocytes were prepared by meshing whole spleens through 70 μm strainers (BD Biosciences) using a syringe plunger. When cytokine production was assessed, CD8$^+$ T cells within splenocytes were stimulated with 1 μg/ml of gp33 peptide in the presence of 10 μg/ml of brefeldin A (Sigma-Aldrich) for 6 h at 37 °C. Fluorophore-conjugated antibodies used for flow cytometry staining were purchased from BD Biosciences (α-CD44 FITC IM7 (1/200); α-CD44 JES6-5H4 (1/100); streptavidin APC (1/500); streptavidin PE-Cy7 (1/500)), eBioscience (Lucerna Chem AG, Luzern, Switzerland) (α-IFNγ PE XMG1.2 (1/100); α-KLRG1 PE-Cy7 2F1 (1/200); α-PD-1 FITC J43 (1/100)), Cell Signalling (Bioconcept, Allschwil, Switzerland) (α-phospho-Akt PE D9E (1/50); α-phospho-NFκB PE 93H1 (1/50); α-phospho-S6 PE D57.2.2E (1/50)) or BioLegend (α-CD28 APC 37.51 (1/200); α-CD122 FITC TM-β1 (1/100); α-CD3ε APC-Cy7 145-2C11 (1/100); α-CD39 AF647 Duha59 (1/100); α-CD44 BV510 IM7 (1/200); α-CD44 PE IM7 (1/200); α-CD45.1 APC A20 (1/200); α-CD45.1 PerCP A20 (1/200); α-CD8 APC-Cy7 53-6.7 (1/100); α-CD8 BV510 53-6.7 (1/200); α-CD8 PerCP 53-6.7 (1/200); α-CD49d biotin R1-2 (1/200); α-CD62L PerCP MEL-14 (1/100); α-CXCR3 BV421 CXCR3-173 (1/100); α-KLRG1 FITC 2F1 (1/200); α-IL-7Rα BV421 A7R34 (1/100); α-LFA-1 PE 2D7 (1/200); α-NKG2D APC CX5 (1/100); α-PD-1 PE-Cy7 29F.1A12 (1/100); α-TNF FITC MP6-XT22 (1/100); streptavidin APC-Cy7 (1/500)). Identification of viable cells was done by fixable near-IR dead cell staining (Life Technologies). Erythrocytes were lysed by ACK lysis buffer treatment for 5 min at room temperature. Surface stainings were performed for 20 min at 4 °C. For cytokine production analysis, cells were further fixed and permeabilized in 2× FACS Lysis Solution (BD Biosciences) with 0.08% Tween 20 (National Diagnostics, Chemie Brunschwig AG, Basel, Switzerland) for 10 min at room temperature, followed by intracellular staining for 30 min at room temperature in the dark. For the assessment of phosphoprotein expression from in vitro activated CD8$^+$ T lymphocytes, surface staining was followed by fixation with pre-heated 4% paraformaldehyde (Sigma-Aldrich) for 10 min at 37 °C, permeabilization with 90% of pre-cooled methanol for 30 min on ice and staining for 40 min at room temperature in the dark. All samples were washed and stored in PBS containing 2% FBS (Omnilab) and 5 mM of EDTA (Sigma-Aldrich) before acquisition. Stained samples were acquired on a FACS LSR II flow cytometer (BD Biosciences) with FACSDiva software. Data analysis was done using FlowJo software (FlowJo Enterprise, version 10.7.1, BD Biosciences). A representative gating strategy is shown in Supplementary Fig. 7.

**Immunofluorescence staining and confocal microscopy.** CD8$^+$ T-cell lymphocytes, previously stimulated in vitro, were washed in PBS and transferred on Poly-L-Lysine (Sigma-Aldrich) treated coverslips, followed by incubation for 30 min at 37 °C. Cells were then fixed with 2% paraformaldehyde (Sigma-Aldrich) for 10 min, permeabilized with 0.1% Triton X (Sigma-Aldrich) for 10 min and blocked in PBS containing 2% bovine serum albumin (GE Healthcare) and 0.01% Tween 20 (National Diagnostics) for 1 h at room temperature. The following antibodies were used to perform immunofluorescence stainings in murine cells: mouse α-β-tubulin (Sigma-Aldrich); α-mouse IgG AF488 (Abcam, Lucerna Chem AG, Luzern, Switzerland); α-CD8 APC (53-6.7, BioLegend); α-T-bet PE (4B10, BioLegend); α-

phospho-S6 PE (D57.2.2E, Cell Signalling); goat α-rabbit IgG (H + L) AF568 (Life Technologies). DAPI (Sigma-Aldrich) was used to detect DNA. Mowiol (Calbiochem, Merck-Millipore, Schaffhausen, Germany) was used as mounting medium. Mitotic cells (late anaphase to cytokinesis) were identified by nuclear morphology, presence of two microtubule organizing centres (MTOCs) and a clear tubulin bridge between two daughter cells. Twenty to thirty Z-stacks were acquired with a Visitron Confocal System or a Nikon A1R confocal system with 100× magnification. Data were analyzed using Volocity software (version 6.3., PerkinElmer). Thresholds for quantification were setup individually for each fluorophore. Asymmetry rates were calculated based on CD8 quantification (volume and fluorescence intensity) in each hemisphere of mitotic cells. Mitoses were considered asymmetric when CD8 enrichment was 1.5-fold greater in one daughter cell in comparison to the other.

**Metabolic flux analysis.** The real-time extracellular acidification rate (ECAR) and oxygen consumption rate (OCR) were measured using a XF 96 extracellular flux analyser (Seahorse Bioscience). 1–2 × 10$^6$ activated CD8$^+$ T cells were washed in RPMI 1640 without sodium bicarbonate, 20 mM glucose, 1% FCS, 2 mM pyruvate and seeded in a XF plate coated with poly-L-lysine (Sigma-Aldrich) at equal densities in corresponding assay medium (for ECAR—XF Assay Medium, pH 7.4; for OCR—XF Assay Medium, pH 7.4 supplemented with 25 mM glucose, 1 mM sodium pyruvate and 2 mM L-glutamine). For measurement of OCR, test compounds were sequentially injected to obtain the following concentrations: 0.75 μM oligomycin, 1 μM FCCP, 1 μM rotenone and 1 μM antimycin A. For quantification of ECAR, 11.1 mM D-glucose, 0.75 μM oligomycin and 100 mM 2-deoxy-D-glucose were used. Two independent experiments were performed with at least three technical replicates per group.

**In vitro survival upon limiting cytokine availability.** Isolated CD8$^+$ T cells (naive or T$_{VM}$) from aged P14 mice were cultured in T-cell medium supplemented with human IL-2, and stimulated on α-CD3 (5 μg/ml) (BioLegend), α-CD28 (5 μg/ml) (BioLegend) and human Fc-ICAM-1 (50 μg/ml) (R&D Biosciences) coated plates for 36 h. When transient mTOR inhibition was conducted, at 12 h post stimulation 20 nM of rapamycin (Santa Cruz) or 1–2 μM of Akt-kinase inhibitor (Sigma-Aldrich) was added to the cell culture. After plate-bound stimulation, cells were harvested, washed in PBS and transferred to uncoated new wells containing limited concentrations of recombinant mouse IL-15 (<1 ng/ml) or a cocktail of recombinant mouse IL-2, IL-7 and IL-15 (<1 ng/ml). Cells were checked for viability by flow cytometry 7 days later.

**Statistical analysis.** To test if data point values were distributed in a Gaussian distribution, Shapiro–Wilk normality test was performed. For statistical analysis, two-tailed Student's $t$ test or non-parametrical analysis Mann–Whitney $U$ tests were performed using GraphPad Prism Software (version 7.0 or 8.2.0). $P$ values were considered significant when <0.05, and exact $P$ values are provided in the figure legends. For animal experiments, sample size varied from 2 to 5 mice per group for each individual experiment.

**Reporting summary.** Further information on research design is available in the Nature Research Reporting Summary linked to this article.

## Data availability
Additional flow cytometry and imaging data supporting the findings are available from the first/corresponding author upon request. Source data for Figs. 1B, 2B–G, 3A, C, 4C, D and 5B–D and Supplementary Figs. 1B, 2 and 4–6 are available in the Source Data file. Source Data are provided with this paper.

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

## Acknowledgements

These authors contributed equally: Isabel Barnstorf, Nicolas S. Baumann, Katharina Pallmer. The authors thank members of the Oxenius, Joller, Barral and Jessberger groups for helpful discussions. This work was supported by ETH Zurich, the Swiss National Science Foundation (grant no. 310030_166078 to A.O., grant no. P2EZP3-188074 to M.Bo.), the Novartis Foundation for Biomedical Research (to A.O. and M.Bo.), the European Union's Horizon 2020 (under the Marie Sklodowska–Curie grant agreement 893676 to M.Bo.) and the Wellcome Trust (103830/Z/14/Z to A.K.S.).

## Author contributions

M.Bo., A.K.S., N.J., Y.B., R.S. and A.O. designed the experiments. M.Bo., N.B., F.G., I.B., N.S.B., K.P., S.B., D.S., M.Ba. and C.W. performed the experiments. N.O. and F.W. provided technical assistance. M.Bo., N.B., F.G. and A.O. analyzed the experiments. M.Bo. and A.O. wrote the manuscript.

## Competing interests

The authors declare no competing interests.
