## [Peer Review File · Nature Communications]

REVIEWER COMMENTS

Reviewer #1 (Remarks to the Author):

Overall this is an interesting study trying to link age-related impairments in ACD to T cell dysfunction. They demonstrate ACD defect nicely, but the rest of the story is a bit muddled. The link between ACD to expansions of IL-7Rahi cells poised to become memory and virtual memory cells is not explained clearly. It seems the most interesting findings with regard to mTOR inhibition were seen in naïve CD8+ T cells and there is no evidence to show this results in changes in the TVM compartment. I think the authors should attempt to link these findings more explicitly.

Specific points:

- 1) It would be helpful to have more on ACD in the introduction.
- 2) >50 weeks is both vague and not what I would consider aged. 100 weeks would be more appropriate. Similarly, why the change from >50 to >70 for the adoptive transfer experiments? That said, differences are seen, so I imagine they might be more stark with even older mice.
- 3) Discrepancy between p-S6 and T-bet asymmetry – what does this mean?
- 4) Is Old + R equivalent to young, or young + R?
- 5) Expression of CD3, CD28 and CD11/CD18 on T cells from these mice? These are the main stimuli for achieving ACD so would be good to see these controls added (or reference to literature).
- 6) Line 119-121: "...higher frequencies of cells committed to a memory fate, defined as KLRG1lo IL-7Rahi..." – while this is true in the lymph nodes, the relative frequency of these cells is still lower compared to the KLRG1hi IL-7Ralo cells (40-70% vs 5-20% of P14 cells). Indeed it appears that KLRG1 expressing cells are unaffected, whereas KLRG1lo cells increase their expression of IL-7Ra. Is the conclusion that existing memory cells are not affected, but instead rapamycin induces a transition of naïve cells towards IL-7Rahi cells that are poised to become long term memory? Would these be conventional or TVM cells and could this be shown? Enumeration of KLRG1lo IL-7Ralo cells might help in understanding the dynamics of this process and what rapamycin is doing.
- 7) Line 122-123: "...exhibited increased cytokine production (TNF; Fig. 2E)" – agreed, but there is no mention of the lack of effect of mTOR on IFN production. What does this mean?
- 8) Why was the metabolic profile of young TVM not assessed? Would one expect these cells to behave like old TVM?
- 9) Fig. 3B: It would be nice to see cumulative data for these experiments and not merely a representative from 2 experiments.
- 10) Fig. 3C: was spare respiratory capacity normalised to young naïve cells? If so, this should be stated.
- 11) Line 156-158: "...a very modest increase in exhausted cells..." – Both PD-1 and CD39 are increased by much more than 2-fold in old compared to young CD44hi cells. This would appear to be a more profound increase than that observed in KLRG1lo IL-7Rahi cells.
- 12) Line 208: "TVM- CD8+ T cells show intrinsically high ACD rates". This has not been formally shown. Is there a statistical analysis comparing naïve vs TVM cells?
- 13) Line 222-223: Fig. S3 shows phosphorylation of downstream targets of mTORC1 and mTORC2, not the effect of Akt1 and Rapamycin on ACD. An inclusion of the double treatment on the effect of ACD would be welcome.

14) Why was the data in figure 5 analysed the way it was. Why compare adoptively transferred naïve cells in young mice to adoptively transferred TVM cells in old mice? These findings need to be explained better.

Reviewer #2 (Remarks to the Author):

Borsa et al present work demonstrating that the capacity for asymmetric cell division changes in CD8 T cells with increasing age. They present a number of novel and exciting ideas. Firstly, they show that ageing leads to a reduction in asymmetric cell division, mainly in the conventional naïve cells, but this age-related deficit can be remedied with treatment with rapamycin. They also show that virtual memory cells have inherently high rates of asymmetric division and are impervious to rapamycin treatment. They suggest that this intrinsic capacity for asymmetric division correlates with increased memory potential and they demonstrate increased secondary responses in an infection model. The authors observations about ACD in ageing and TVM cells reflect a new, previously unexplored facet of T cell biology.

The manuscript is well written, the experimental work appears to be well performed and thorough and the authors' analysis, description and interpretation of the dataset is fair and accurate. As a result, I have a few major but mainly minor comments below although I would like the authors to clarify two key points to ensure I have interpreted the work correctly- were the P14 mice Rag deficient and were the cells used in Figure 3C previously stimulated with the ACD conditions?

Major:

The paper uses P14 mice with a polyclonal stimulation and an LCMV challenge model. Given that signalling through endogenous, modestly self-reactive TCRs is a likely factor in TVM cell development and retention with age, the use of a TCR transgenic system has a few implications for the paper:

-I assume based on the CD44/CD49d expression profile (and my comments assume) that these mice are Rag deficient? In any case, it would be helpful for the reader to specify the status of Rag expression in the Materials and Methods section.

-Given that self-stimulation through the endogenous TCR is thought to have a major impact on TVM cell generation, maintenance and ageing, I think it would be optimal if the authors validated key parameters with young and old TCR polyclonal cells. For example, are the same ACD rates seen with young and old WT conventional naïve and virtual memory cells? Are the same proliferation kinetics seen with young and old WT conventional naïve and virtual memory cells? If the mice used are Rag sufficient and express endogenous TCR, this is less of a concern but the authors should consider including it and discuss it.

Panel 3C needs a little more information- it says in the results text at line 184 that the cells are naïve but at line 189 is discusses activated TVM cells. It says in the legend that the cells are measured under basal conditions but it says in the methods that cells are activated- so are the cells that are used in the Seahorse assay immediately post-stimulation with the ACD protocol? If so, they TVM progeny that are no longer naïve so the language should be made consistent with this. And if so, how long have they been stimulated for? The methods also don't say which machine was used for analysis- was it an XF24? How many T cells were plated per condition? It would be helpful to have data for young TVM cells as well, but I understand that the population is at a very low frequency in P14 animals and this may not be feasible.

The authors should also interpret the metabolic data a bit more explicitly in the section from line 180-192 and in the discussion. Can the authors point to specific metabolic outcomes that they are measuring and describe and reference how outcomes have been linked to specific T cell functions (such as cell cycling speed?)? Simplistically, my interpretation would be that, in stimulated aged TVM cells, the retention of OCR may support cell survival and memory potential and the glycolytic capacity may support rapid proliferation, but the authors don't state this (or another interpretation) clearly. The

SRC data should also be interpreted more explicitly. Also at line 312, the authors say that old TVM cells have a mixed effector memory metabolic phenotype- could the authors expand on this characterisation?

In addition, another way to display the metabolic data in a simplified plot could be to use just the MitoStress assay data- the authors should have OCR and matched ECAR data for this. If they plot OCR vs ECAR for a basal respiration timepoint and OCR vs ECAR for a maximal respiration timepoint, they can connect the two points for each cell type and this will give them the trajectory for cellular metabolism under stress. The authors can divide the plot into quadrants- OXPHOS-reliant, glycolysis-reliant, mixed metabolism and quiescent (see the Seahorse Energy Phenotype test). This plot should show that old TVM cells have features that are in between young and old naïve cells- retention of OCR capacity but robust glycolytic capacity at basal respiration. However, under stress (at maximal respiration), the old TVM (and the young naïve cells) should shift toward a more glycolytic profile.

At lines 188-192, the concept of TVM cells as metabolically poised for rapid division AND “slow dividers” is hard to reconcile as once they do start dividing, they are explosive. It would be more accurate to say their “time to first division” is extended (noting that they are more synchronous in division progression when they initiate division, which I think is really interesting) but they have more metabolic flexibility to support this proliferation. Metabolic flexibility may be a more relevant concept here rather than capacity.

Figure 5 is important data for understanding the contribution of TVM cells to memory responses- however, it uses cell number recovered after a secondary response as a measure. There are two memory functions that may be determinants of this measure i) the number of cells that engraft after transfer and survive for 30 days until challenge and ii) the fold expansion of these cells during the secondary response. Importantly, old CD8 T cells in general and old TVM cells in particular seem to have a survival advantage (Davenport et al, JI, 2019; Quinn et al, Cell Rep, 2018), which may explain why old TVM cells have a much more robust response- they may engraft and survive at the higher rate and there would therefore be more of them to respond to the challenge. Lee et al, PNAS, 2013 also suggests that TVM cell progeny adopt a TCM phenotype more readily, which would support increased potential for higher fold expansion. The authors should consider dissecting these two functions experimentally- by doing transfers (of more cells to aid recovery) and characterising the relative engraftment of each population 30 days after transfer, and by performing restimulation of stimulated and rested progeny and characterising their fold expansion (could be done in vitro). Experimentally, these points- the memory potential and restimulation capacity of TVM progeny- are critical outstanding questions in the field- essentially, are TVM progeny useful for generating memory populations? However, I don't think it should impede the publication of this work if it can't be done, especially with regard to potential COVID restrictions. In this case, it would at least be helpful to discuss the dual contribution of engraftment/survival and re-expansion capacity to the overall number of cells recovered after challenge in the discussion paragraph from lines 328-334.

Similarly, on line 348, it isn't quite accurate to say that the authors saw “increased expansion” after challenge as fold expansion from the point of challenge can't be calculated without knowing the numbers of cells present at the time of challenge- it would be better here to “increased recovery of transferred cells”.

The findings in this paper could explain a result in Lee et al, PNAS, 2013 that has long puzzled me. Using the Listeria model, this study found that TVM cells preferentially differentiated into short-lived effector cells AND into central memory cells, as compared to conventional naïve cells. Could this more distinct polarisation be related to the increased ACD in TVM cells (and could the increased secondary responses be a function of increased TCM generation)? Perhaps the authors could comment on that in the discussion?

More as a comment- could there be a survivorship bias with analysing old TVM cells at 72 hours after division? Given that old TVM cells don't divide as well in vitro, if you are just assessing asymmetric

division in those that divide, you are missing the impact of stimulation on those that don't divide and die. So is the ACD data for old TVM cells relevant for a subset of old TVM cells that have retained robust proliferative function? And this would be another contributing factor for why the characteristics of conventional naïve cells dominate in figure 1- because the TVM cells don't proliferate as well and are therefore not included in the dataset. Might be worth noting briefly?

Minor:

Line 38: Phrasing doesn't quite make sense. I suggest "...particularly with regards to the formation"

Line 86: Should be identifiable

Line 119: Should be "Phenotypic changes..."

Line 129 and Line 234: Should be "in the presence or not of"

Line 198: Delete "prolife"

Line 201: Should be "...was assessed 72h..."

Line 215: Should be "previously"

Line 223 refers to Figure S3 as demonstrating ACD rates with combined Akti and Rapamycin treatment but it just shows phosphorylation of S6 and NFkB- is there an ACD rate figure missing? Or should this refer to Figure 4C?

Line 309: Delete "increases markedly"

Line 314-318: Consider refining this sentence or breaking into two- doesn't quite make sense.

Figure 3A and Figure 5 have data pooled from 2 independent experiments- please show the datapoints. It is hard to assess impact of pooling the data without the datapoints.

Figure 3B should be quantified with a companion bar chart or similar graphical display with biological replicates.

Figure 3C: the dotted lines are hard to see- could the authors use solid lines?

Figure 3C: Do the authors mean "glycolytic capacity" rather than just "glycolytic".

Figure 4C: The key should say "symmetric"

Figure S3: Should be "mean fluorescence intensity"

Signed review: Kylie Quinn

Reviewer #3 (Remarks to the Author):

In this paper, Borsa et al follow up on their previous observations that rapamycin can increase asymmetric cell division (ACD) in CD8 T cells. Here they show a) that CD8 T cells from aged mice display less ACD upon activation than do CD8 T cells from young mice, b) that rapamycin increases both ACD and memory generation in total or naïve CD8 T cells from aged mice, c) that virtual memory CD8 T cells (Tvm) display ACD that is not affected by rapamycin or AKTi. In general, the data look robust, although there are a few issues and concerns. These are as follows.

1. Other studies have shown that the CD8low population, following first division, is enriched in cells that become memory cells. Based on this, the authors wish to conclude that, by increasing ACD with rapamycin treatment, they increase the numbers of these CD8low cells and therefore memory. There is, however, another possibility: CD8high cells produced in the first division do not normally contribute to memory, but do so when treated with rapamycin (as shown by Jonathan Powell and others). Unlike most other studies of ACD in CD8 T cells, the authors do not examine the relative contributions of the CD8hi versus CD8lo first division T cells. Without this, it is difficult to resolve these possibilities. Thus, it will be important to do these sorts with or without rapamycin treatment, and assess contribution to memory, both for naïve and Tvm cells. Without such experiments, the conclusion, as stated in the title, is only supported by correlation.

2. While numbers of P14 T cells are enumerated in recipient, challenged mice in Figures 2 and 5, only percentages are provided for subsets and cytokine production, necessitating recalculation by the reader (which can only be estimated, and significance is not addressable). Thus, when % Te, Tem, and Tcm (or TNF, or IFNg producing cells) show no difference (or some difference), what we really

need to know is how many of these there are, and if the differences are significant. The same will apply to the experiments requested in #1.

3. The values for % TNF producing cells in Figure 2 and Figure 5 are strikingly different. Such experimental variation is troubling for any conclusions.

4. Often, experiments are only done twice, and in most cases do we know how many individual animals were used. For example, in Figure 3B, one cell division of two is shown. We certainly need to know how general the effects are.

5. The results in Figure 3 are interesting (and the Seahorse experiments are excellent), but not really integrated into the rest of the paper (other than to cite results from others). How does rapamycin treatment affect these differences? Are the effects related to ACD? (That is, are the CD8hi or CD8lo cells responsible for these differences?)

6. A critical experiment is shown in Figure 5B, which is the only experiment that shows that numbers of old P14 cells are decreased upon challenge, and that this increases with rapamycin treatment to levels seen with young, rapamycin-treated cells. This experiment should be re-drawn to highlight this effect, which after all is the conclusion in the title of the paper. Further, as indicated in #1, the numbers rather than percentages for the subsets should be presented to highlight the effects. (see also #7).

7. A major basis for the studies herein is that T cells from aged mice generate less memory. They do not actually show us this point (in Figure 2 there is no comparison with young T cells). If, indeed, rapamycin increases memory to the level seen in young T cells (we do not know this—it is only shown for young, naïve T cells in Figure 5B), we may be able to attribute this effect to the increase seen in the naïve but not Tvm subset. For this, however, we need a direct comparison in which total, naïve, and Tvm are compared in an experiment such as that in Figure 5B.

REVIEWER COMMENTS

Reviewer #1 (Remarks to the Author):

Overall this is an interesting study trying to link age-related impairments in ACD to T cell dysfunction. They demonstrate ACD defect nicely, but the rest of the story is a bit muddled. The link between ACD to expansions of IL-7Rahi cells poised to become memory and virtual memory cells is not explained clearly. It seems the most interesting findings with regard to mTOR inhibition were seen in naïve CD8+ T cells and there is no evidence to show this results in changes in the TVM compartment. I think the authors should attempt to link these findings more explicitly.

We would like to thank the reviewer for the constructive criticism to our story. It seems we were not clear enough while representing and discussing our data, something we have now addressed in the revised version of the manuscript. We did not suggest any causal link/correlation between IL7Ra expansions to virtual memory development. The fact TVMs did not have ACD modulated while naïve CD8 T cells (from aged mice) did, was indeed surprising, but also the reason for our interpretation that transient mTOR inhibition might improve naïve T cells responses in the elderly, while TVMs might be an adaptation (and not maladaptive, as other studies suggest) to immunosenescence.

Specific points:

1) It would be helpful to have more on ACD in the introduction.

We added new information about ACD in the introduction (highlighted text).

2) >50 weeks is both vague and not what I would consider aged. 100 weeks would be more appropriate. Similarly, why the change from >50 to >70 for the adoptive transfer experiments? That said, differences are seen, so I imagine they might be more stark with even older mice.

100 weeks (2 years) is the maximum limit of age to keep an animal for an ageing experiment in most countries following FELASA regulations. Many animals develop age-related co-morbidities while ageing, which is also an important ethical issue. The aged mice we used for the experiments were not purchased, but aged in our laboratory, where we could witness a steep drop in survival after 18 months of age. When we performed the first experiments, our aim was to evaluate whether age would have an impact in the ability of CD8 T cells to undergo asymmetric cell division, which could be confirmed even with mice maybe not considered aged by the reviewer. We agree that differences could be stronger using older animals, but as the results using “middle-aged” mice were confirmed by subset-specific experiments done later on in animals with more advanced age, we believe it would be ethically reasonable not to repeat them just to follow an age-threshold definition. In fact, following the Jackson Laboratory guidelines, mice between 10 and 15-months-old can be considered middle-aged, and above that age can be called aged (<https://www.jax.org/news-and-insights/jax-blog/2017/november/when-are-mice-considered-old>). This means the first experiments were done in middle-aged to aged mice (we have changed the text accordingly to explicitly highlight that), while further experiments were done with aged mice.

3) Discrepancy between p-S6 and T-bet asymmetry – what does this mean?

Figure S1 shows that there is no discrepancy between p-S6 and T-bet asymmetry – both show a trend (statistically significant for T-bet) to be co-inherited with CD8 upon ACD, which supports previous

observations (Chang et al., 2011, Immunity; Verbist et al., 2016, Nature; Pollizzi et al., 2016, Nature Immunology).

4) Is Old + R equivalent to young, or young + R?

We believe the reviewer is not considering the TVM compartment in this question, as these cells do not have ACD (or memory potential) being modulated by transient rapamycin inhibition.

Concerning ACD rates: in bulk CD8 T cells, old+R is equivalent to young (Fig.1); in naïve CD8 T cells, old+R is equivalent to young+R (Fig. 4).

Concerning the re-expansion potential (as a measure of memory potential): in naïve cells, old+R is at least equivalent to young (Fig.5).

5) Expression of CD3, CD28 and CD11/CD18 on T cells from these mice? These are the main stimuli for achieving ACD so would be good to see these controls added (or reference to literature).

We analysed the expression of the mentioned markers in the blood of young and aged P14 mice. The data can be found below (Fig. A). We now refer to this in the text and show it as an additional supplementary figure in the manuscript (Figure S4). Overall, the expression of CD3 and CD28 was slightly lower in CD44^{hi} (T_{VM} compartment) cells, while LFA-1 (CD11/CD18) was upregulated in this compartment. As ICAM-1/LFA-1 binding is a requirement for ACD (Chang et al., 2007, Science; Borsa et al., 2019, Science Immunology), we may speculate that perhaps increased LFA-1 expression could contribute to enhanced ACD in T_{VM} cells. This hypothesis was added to the discussion.

Figure A. Peripheral blood of P14 young and old mice was taken for phenotypical characterization of immunological synapse molecules expression in CD8 T cells. Depicted are representative plots of CD8 T cells showing co-expression of CD44 and CD3 (upper panel), LFA-1 (middle panel) or CD28 (lower panel).

6) Line 119-121: "...higher frequencies of cells committed to a memory fate, defined as KLRG1^{lo} IL-7Ra^{hi}..." – while this is true in the lymph nodes, the relative frequency of these cells is still lower compared to the KLRG1^{hi} IL-7Ra^{lo} cells (40-70% vs 5-20% of P14 cells). Indeed it appears that KLRG1^{hi} expressing cells are unaffected, whereas KLRG1^{lo} cells increase their expression of IL-7Ra. Is the conclusion that existing memory cells are not affected, but instead rapamycin induces a transition of naïve cells towards IL-7Ra^{hi} cells that are poised to become long term memory? Would these be conventional or TVM cells and could this be shown? Enumeration of KLRG1^{lo} IL-7Ra^{lo} cells might help in understanding the dynamics of this process and what rapamycin is doing.

This question refers to Fig.2. Following the reviewer's suggestion, cell numbers of KLRG1^{hi} vs IL7Ra^{hi} cells were added for clarification.

Asymmetric cell division has been widely reported to contribute to generation of diversity, which in the context of CD8 T cells relies on stemness, a feature of memory cells. In immune responses, cells that commit to an effector fate outnumber cells that retain memory potential. Effector CD8 T cells expand very quickly, which is achieved by symmetric cell division (Borsa et al., 2019, Science Immunology), being independent of higher or lower ACD rates. ACD is particularly critical in the early steps after antigen recognition, where it has been shown to benefit the generation of daughter cells with a memory-like phenotype (Chang et al., 2011, Immunity; Verbist et al., 2016, Nature; Pollizzi et al., 2016, Nature Immunology; Borsa et al., 2019, Science Immunology). Transient rapamycin treatment, from 12h to 36h post-stimulation of CD8 T cells in vitro - and prior to first mitosis, is able to improve the capacity of CD8 T cells to divide asymmetrically, which is beneficial for the establishment of memory potential in one of the daughter cells, but not detrimental to effector differentiation. Thus, we were not surprised to see almost unchanged frequencies of KLRG1^{hi} cells when old versus old+R conditions were compared. Furthermore, the higher frequencies of IL7Ra^{hi} cells were expected, based on the higher ACD rates exhibited in the old+R group.

Considering all the functional analyses were done in TCR transgenic P14 cells, which express a TCR specific for a peptide of the glycoprotein of LCMV (gp₃₃₋₄₁), and that these cells were re-challenged upon LCMV infection, memory cells rising from this setup are cognate-antigen driven, which excludes the possibility of them being T_{VM} cells.

7) Line 122-123: "...exhibited increased cytokine production (TNF; Fig. 2E)" – agreed, but there is no mention of the lack of effect of mTOR on IFN production. What does this mean?

We do not want to make a big point about differences in per cell cytokine production. We previously reported that transient mTOR inhibition (and enforced ACD) overall does not change the ability of T cells to produce cytokines, but increases the frequencies of cells that survive for a long period in absence of antigenic stimulation (memory-like daughter cells) (Borsa et al., 2019, Science Immunology). This leads to differences in numbers (of cytokine producing cells), which was added in the figure and further emphasized in the text.

8) Why was the metabolic profile of young TVM not assessed? Would one expect these cells to behave like old TVM?

The methods we had available required high cell numbers for the metabolic profiling. As T_{VM} cells are very rare in young mice (approximately up to 5% of total CD8 T cells), this would require a very high number of animals to be culled. The aim of our metabolic profiling was to better characterize T_{VM} cells in order to understand whether they are an adaptive or maladaptive mechanism upon ageing, which was the main reason why we focused on the T_{VM} compartment of aged hosts for this specific experiment. For experiments requiring lower cell numbers, such as in adoptive transfer experiments, we also included the T_{VM} cells of the groups of young mice.

9) Fig. 3B: It would be nice to see cumulative data for these experiments and not merely a representative from 2 experiments.

We would like to apologize, but after checking the archived data from our second experiment, we realised we had slightly different time points (the later time point was at day 4 instead of day 3, being the reason why we unfortunately are not able to show cumulative data). However, our aim in Figure 3B was to show that the previously reported proliferation defect observed in T_{VM} cells is restricted to earlier time points (Quinn et al., 2018, Cell Reports, confirmed by us). At later time points, we observed in both experiments that T_{VM} cells exhibited higher proliferation rates (Figure 3B, manuscript; Figure B, below, upper panel), suggesting they have a unique cell division kinetics, as discussed in the manuscript.

To further confirm this phenotype, as we did not have any P14 aged mice available as a consequence of the pandemic, we performed a similar experiment using naïve and T_{VM} cells from young and aged wild type B6 cells (Figure B, below, lower panel). Similar to what we observed when P14 cells were analysed, naïve CD8⁺ T cells from young and aged mice showed similar proliferation profiles. Furthermore, T_{VM} cells took longer to undergo their first mitosis, but at 72h post-activation had undergone similar numbers of cell cycles in comparison to their naïve counterparts, being the T_{VM} cells from aged mice the ones that divided less.

Figure B. Histograms depicting proliferation of purified naïve or T_{VM} $CD8^+$ T cells from young or aged mice (CTV-labelled, stimulated on α -CD3, α -CD28 and human Fc-ICAM-1 coated wells). Upper panel: P14 cells. After 38h, cells were transferred to uncoated wells for further 58h in medium containing IL-2, IL-7, and IL-15, until their proliferation kinetics was accessed 94h post-stimulation (left). Another similar experiment, lacking the young T_{VM} group, further confirmed the proliferation capacity of T_{VM} cells 5 days post-activation (right). Lower panel: $CD8^+$ T cells from B6 mice: After 40h, cells were transferred to uncoated wells for further 32h in medium containing IL-2, IL-7, and IL-15, until their proliferation kinetics was accessed 32h post-stimulation. Pre-gating: singlets, near-IR live/dead $^-$, $CD8^+$, $CD44^+$.

10) Fig. 3C: was spare respiratory capacity normalised to young naïve cells? If so, this should be stated.

No, it is not. We would like to emphasize that the metabolic profiling was done with activated naïve cells and not untouched naïve cells. As this question was not exclusively raised by this reviewer, we are aware we need to explicitly address it in the text. We would like to thank the reviewer for bringing this to our attention.

11) Line 156-158: "...a very modest increase in exhausted cells..." – Both PD-1 and CD39 are increased by much more than 2-fold in old compared to young $CD44^{hi}$ cells. This would appear to be a more profound increase than that observed in KLRG1 lo IL-7Rahi cells.

These are two different experiments and readouts. The increased percentages of PD-1^{hi} and CD39^{hi} (markers normally associated with activation/exhaustion) expressing cells within the CD44^{hi} compartment were observed *ex vivo* in the blood of aged P14 mice. Despite the “significant” fold-change observed in comparison to their young counterparts, these cells represented between 1 and 5% of the total CD8 T cell compartment, and around 10% of the CD44^{hi} compartment. We have rephrased the sentence so that it is more accurate. Our intention was to highlight that these cells are found in very small frequencies in aged P14 mice, and that these data do not support the interpretation that the T_{VM} compartment exhibits overall an exhausted phenotype. Importantly, this observation corroborated data from Quinn et al., 2018, Cell Reports (Reviewer 2).

12) Line 208: “TVM⁻ CD8⁺ T cells show intrinsically high ACD rates”. This has not been formally shown. Is there a statistical analysis comparing naïve vs TVM cells?

The data representing the “intrinsically” high ACD rates is found Fig. 4, where it is clear that T_{VM} cells from aged mice divide more asymmetrically than their naïve counterparts (despite the similar ACD frequencies based on the chosen threshold of $(P1-P2)/(P1+P2) \geq 0.2$). As this is indeed not directly addressed, we would like to thank the reviewer for the suggestion. We added a graph to Fig. 4 comparing CD8 segregation in naïve and T_{VM} cells from young and aged P14 animals when not submitted to any pharmacological intervention (including stats).

13) Line 222-223: Fig. S3 shows phosphorylation of downstream targets of mTORC1 and mTORC2, not the effect of Akt1 and Rapamycin on ACD. An inclusion of the double treatment on the effect of ACD would be welcome.

We are aware that this experiment does not directly address ACD modulation. However, it suggests T_{VM} cells are refractory to ACD modulation by rapamycin or Akt inhibitor because their mTORC1 and specially their mTORC2 activities are higher than in naïve cells. As rapamycin is also able to inhibit downstream targets of mTORC2, we interpreted that constitutive mTORC2 activity can be the reason for both the lack of ACD modulation in T_{VM} cells and their efficient glycolytic metabolism. Therefore, in our view, double treatment would not add further insights, since rapamycin already targets both mTORC1 and mTORC2 (which would be the aim of a simultaneous rapamycin + Akt inhibitor treatment).

14) Why was the data in figure 5 analysed the way it was. Why compare adoptively transferred naïve cells in young mice to adoptively transferred TVM cells in old mice? These findings need to be explained better.

All cells (from young or aged P14 mice) were transferred to young hosts. Thus, we addressed cell intrinsic features of immunosenescence. Interestingly, a recent paper shows a strong impact of T cells in overall age-related multimorbidity (Mittelbrunn et al, 2020, Science, DOI: 10.1126/science.aax0860), corroborating the T cell-autonomous role in senescence.

Reviewer #2 (Remarks to the Author):

Borsa et al present work demonstrating that the capacity for asymmetric cell division changes in CD8 T cells with increasing age. They present a number of novel and exciting ideas. Firstly, they show that ageing leads to a reduction in asymmetric cell division, mainly in the conventional naïve cells, but this

age-related deficit can be remedied with treatment with rapamycin. They also show that virtual memory cells have inherently high rates of asymmetric division and are impervious to rapamycin treatment. They suggest that this intrinsic capacity for asymmetric division correlates with increased memory potential and they demonstrate increased secondary responses in an infection model. The authors observations about ACD in ageing and TVM cells reflect a new, previously unexplored facet of T cell biology.

We would like to thank the reviewer for the extremely constructive feedback on our manuscript and the incredible insights given (including data interpretation and hypotheses raised), which was very valuable to improve the quality of our manuscript.

The manuscript is well written, the experimental work appears to be well performed and thorough and the authors' analysis, description and interpretation of the dataset is fair and accurate. As a result, I have a few major but mainly minor comments below although I would like the authors to clarify two key points to ensure I have interpreted the work correctly- were the P14 mice Rag deficient and were the cells used in Figure 3C previously stimulated with the ACD conditions?

Major:

The paper uses P14 mice with a polyclonal stimulation and an LCMV challenge model. Given that signalling through endogenous, modestly self-reactive TCRs is a likely factor in TVM cell development and retention with age, the use of a TCR transgenic system has a few implications for the paper: -I assume based on the CD44/CD49d expression profile (and my comments assume) that these mice are Rag deficient? In any case, it would be helpful for the reader to specify the status of Rag expression in the Materials and Methods section.

P14 mice did not have a Rag-deficient background. However, even in a Rag-deficient background, this would not alleviate potential (low-level) self-reactivity as the P14 TCR was originally positively selected in the thymus by a low affinity self-peptide.

We had analysed the frequencies of T_{VM} cells in the peripheral blood of B6 and P14 mice housed in the same room of our animal facility and could not detect any huge discrepancies in the frequencies of naïve, (CD44^{lo}), T_{VM} (CD44^{hi} CD49d^{lo}) and true memory CD8 T cells (CD44^{hi} CD49d^{hi}) (Figure C).

Figure C. Peripheral blood of B6/P14 young and old mice was taken for phenotypical characterization of CD8 T cells compartments in unimmunized animals. Depicted are representative plots of CD8 T cells showing expression of CD44 and CD49d.

-Given that self-stimulation through the endogenous TCR is thought to have a major impact on T_{VM} cell generation, maintenance and ageing, I think it would be optimal if the authors validated key parameters with young and old TCR polyclonal cells. For example, are the same ACD rates seen with young and old WT conventional naïve and virtual memory cells? Are the same proliferation kinetics seen with young and old WT conventional naïve and virtual memory cells? If the mice used are Rag sufficient and express endogenous TCR, this is less of a concern but the authors should consider including it and discuss it.

As indicated by the reviewer, the suggested experiments would be necessary in case mice used would have been Rag-deficient, which was not the case. However, after a discussion with the editor we agreed it would be relevant to show whether in conventional wt B6 mice:

1. T_{VM} cells would also show intrinsically higher ACD rates,
2. naïve cells from aged animals could have their ACD rates enforced by transient mTOR inhibition,
3. T_{VM} cells would also be refractory to ACD modulation as shown for P14 cells.

The results from this experiment (Figure D) were added as a supplementary figure (Figure S5) in the manuscript.

Figure D. Upper panel: Confocal images from murine naïve and TVM wt B6 CD8⁺ T cells fixed 36-40h after in vitro stimulation on α -CD3, α -CD28 and human Fc-ICAM-1 coated wells in presence or not of transient mTOR inhibition, which was achieved by treatment with rapamycin (R) (20nM). Cells were exposed to drug treatment from 12h post-activation until fixation for confocal microscopy analysis. Lower panel: CD8 and T-bet asymmetry rates in naïve and TVM CD8⁺ T cells isolated from young or old wt B6 mice. Data are shown as mean \pm SEM. Pooled data from 2 technical replicates (2 to 5 pooled animals). Statistical analysis was performed using the unpaired two-tailed Student's t test. *P < 0.05; ****P < 0.0001.

Panel 3C needs a little more information- it says in the results text at line 184 that the cells are naïve but at line 189 is discusses activated TVM cells. It says in the legend that the cells are measured under

basal conditions but it says in the methods that cells are activated- so are the cells that are used in the Seahorse assay immediately post-stimulation with the ACD protocol? If so, they TVM progeny that are no longer naïve so the language should be made consistent with this. And if so, how long have they been stimulated for? The methods also don't say which machine was used for analysis- was it an XF24? How many T cells were plated per condition? It would be helpful to have data for young TVM cells as well, but I understand that the population is at a very low frequency in P14 animals and this may not be feasible.

The reviewer is correct, metabolic analysis was done with progenies of naïve or T_{VM} cells at the same time point cells were analysed by confocal microscopy for ACD. The text was changed accordingly.

Concerning the machine and numbers of cells analysed, an XF96 Extracellular Flux Analyser was used (information was added in the Methods session), and $1-2 \times 10^6$ CD8⁺ T cells were analysed (progenies of activated cells)

The authors should also interpret the metabolic data a bit more explicitly in the section from line 180-192 and in the discussion. Can the authors point to specific metabolic outcomes that they are measuring and describe and reference how outcomes have been linked to specific T cell functions (such as cell cycling speed)? Simplistically, my interpretation would be that, in stimulated aged TVM cells, the retention of OCR may support cell survival and memory potential and the glycolytic capacity may support rapid proliferation, but the authors don't state this (or another interpretation) clearly. The SRC data should also be interpreted more explicitly. Also at line 312, the authors say that old TVM cells have a mixed effector memory metabolic phenotype- could the authors expand on this characterisation?

We thank the reviewer for the suggestions, and we have changed the text accordingly.

In addition, another way to display the metabolic data in a simplified plot could be to use just the MitoStress assay data- the authors should have OCR and matched ECAR data for this. If they plot OCR vs ECAR for a basal respiration timepoint and OCR vs ECAR for a maximal respiration timepoint, they can connect the two points for each cell type and this will give them the trajectory for cellular metabolism under stress. The authors can divide the plot into quadrants- OXPHOS-reliant, glycolysis-reliant, mixed metabolism and quiescent (see the Seahorse Energy Phenotype test). This plot should show that old TVM cells have features that are in between young and old naïve cells- retention of OCR capacity but robust glycolytic capacity at basal respiration. However, under stress (at maximal respiration), the old TVM (and the young naïve cells) should shift toward a more glycolytic profile.

As we do not have matched OCR and ECAR data, we are unable to represent the data in the proposed format. Each type of metabolic readout was done with different samples. However, we would like to thank the reviewer for the suggestion, and we will definitely consider each in further studies. Furthermore, and as previously mentioned, we have changed the text to more explicitly discuss the unique metabolic profile of T_{VM} activated cells/progenies.

At lines 188-192, the concept of TVM cells as metabolically poised for rapid division AND "slow dividers" is hard to reconcile as once they do start dividing, they are explosive. It would be more accurate to say their "time to first division" is extended (noting that they are more synchronous in division progression when they initiate division, which I think is really interesting) but they have more metabolic flexibility to support this proliferation. Metabolic flexibility may be a more relevant concept here rather than capacity.

We thank the reviewer for the suggestions, and we have changed the text accordingly.

Figure 5 is important data for understanding the contribution of TVM cells to memory responses- however, it uses cell number recovered after a secondary response as a measure. There are two memory functions that may be determinants of this measure i) the number of cells that engraft after transfer and survive for 30 days until challenge and ii) the fold expansion of these cells during the secondary response. Importantly, old CD8 T cells in general and old TVM cells in particular seem to have a survival advantage (Davenport et al, JI, 2019; Quinn et al, Cell Rep, 2018), which may explain why old TVM cells have a much more robust response- they may engraft and survive at the higher rate and there would therefore be more of them to respond to the challenge. Lee et al, PNAS, 2013 also suggests that TVM cell progeny adopt a TCM phenotype more readily, which would support increased potential for higher fold expansion. The authors should consider dissecting these two functions experimentally- by doing transfers (of more cells to aid recovery) and characterising the relative engraftment of each population 30 days after transfer, and by performing restimulation of stimulated and rested progeny and characterising their fold expansion (could be done *in vitro*). Experimentally, these points- the memory potential and restimulation capacity of TVM progeny- are critical outstanding questions in the field- essentially, are TVM progeny useful for generating memory populations? However, I don't think it should impede the publication of this work if it can't be done, especially with regard to potential COVID restrictions. In this case, it would at least be helpful to discuss the dual contribution of engraftment/survival and re-expansion capacity to the overall number of cells recovered after challenge in the discussion paragraph from lines 328-334.

Performing these long-term experiments would now be extremely challenging/not feasible, and we are very grateful for the understanding of this reviewer. However, we agree that addressing survival would be a very relevant point. While we do not have the data suggested by the reviewer, we have done *in vitro* experiments that suggest that T_{VM} cells are indeed better survivors, which would contribute to their higher frequencies and numbers upon secondary responses (Figure E). We cultured naïve and T_{VM} cells from aged P14 animals in limiting cytokine concentrations, and observed T_{VM} cells show better survival (measured by numbers of viable cells) in these conditions. In our previous publication, we have seen that cells that survive better in limiting IL-15 conditions *in vitro* ($CD8^lo$ in comparison to $CD8^{hi}$) also survive better after adoptive transfer *in vivo* (Borsa et al., 2019, Science Immunology). Thus, we can speculate that it is very likely the progenies from old T_{VM} 's survive better than the ones from old naïve cells *in vivo* as well. We added this to the discussion of the manuscript and the figure below as a supplementary figure.

Figure E. Isolated naïve or T_{VM} CD8⁺ T cells from aged mice were cultured in T cell medium supplemented with human IL-2, and stimulated on α -CD3 (5 μ g/ml) (BioLegend), α -CD28 (5 μ g/ml) (BioLegend) coated plates in presence or not of plate-bound human Fc-ICAM-1 (50 μ g/ml) (R&D Biosciences) for 30-36 hours. After plate-bound stimulation, cells were harvested, washed in PBS, and transferred to uncoated new wells containing limited concentrations of recombinant mouse IL-15 or a cocktail containing IL-2, IL-7, IL-15 (<1 ng/ml). Cells were analyzed for viability by flow cytometry 7 days later. Data representative of 1 out of 2 experiments with technical duplicates.

Similarly, on line 348, it isn't quite accurate to say that the authors saw "increased expansion" after challenge as fold expansion from the point of challenge can't be calculated without knowing the numbers of cells present at the time of challenge- it would be better here to "increased recovery of transferred cells".

We agree with the reviewer and we have changed the text accordingly.

The findings in this paper could explain a result in Lee et al, PNAS, 2013 that has long puzzled me. Using the Listeria model, this study found that TVM cells preferentially differentiated into short-lived effector cells AND into central memory cells, as compared to conventional naïve cells. Could this more distinct polarisation be related to the increased ACD in TVM cells (and could the increased secondary responses be a function of increased TCM generation)? Perhaps the authors could comment on that in the discussion?

That's a very interesting point and we have discussed this hypothesis in the text (line 177-179).

More as a comment- could there be a survivorship bias with analysing old TVM cells at 72 hours after division? Given that old TVM cells don't divide as well in vitro, if you are just assessing asymmetric division in those that divide, you are missing the impact of stimulation on those that don't divide and die. So is the ACD data for old TVM cells relevant for a subset of old TVM cells that have retained robust proliferative function? And this would be another contributing factor for why the characteristics of conventional naïve cells dominate in figure 1- because the TVM cells don't proliferate as well and are therefore not included in the dataset. Might be worth noting briefly?

We agree with the reviewer that the ACD analysis might focus on a small percentage of T_{VM} cells that can undergo division at this early stage (36h post-stimulation). Thus, it is indeed likely that the mitoses from T_{VM} cells are under-represented in Figure 1 (bulk CD8 T cells). We have added a brief discussion about it in the text as suggested (line 225-227).

Minor: We thank the reviewer for the thorough review, and we have changed the text accordingly.

Line 38: Phrasing doesn't quite make sense. I suggest "...particularly with regards to the formation"

Line 86: Should be identifiable

Line 119: Should be "Phenotypic changes..."

Line 129 and Line 234: Should be "in the presence or not of"

Line 198: Delete "prolife"

Line 201: Should be "...was assessed 72h..."

Line 215: Should be "previously"

Line 223 refers to Figure S3 as demonstrating ACD rates with combined Akti and Rapamycin treatment but it just shows phosphorylation of S6 and NFkB- is there an ACD rate figure missing? Or should this refer to Figure 4C?

Line 309: Delete "increases markedly"

Line 314-318: Consider refining this sentence or breaking into two- doesn't quite make sense.

Figure 3A and Figure 5 have data pooled from 2 independent experiments- please show the datapoints. It is hard to assess impact of pooling the data without the datapoints.

Figure 3B should be quantified with a companion bar chart or similar graphical display with biological replicates.

Figure 3C: the dotted lines are hard to see- could the authors use solid lines?

Figure 3C: Do the authors mean "glycolytic capacity" rather than just "glycolytic".

Figure 4C: The key should say "symmetric"

Figure S3: Should be "mean fluorescence intensity"

Signed review: Kylie Quinn

Reviewer #3 (Remarks to the Author):

In this paper, Borsa et al follow up on their previous observations that rapamycin can increase asymmetric cell division (ACD) in CD8 T cells. Here they show a) that CD8 T cells from aged mice display less ACD upon activation than do CD8 T cells from young mice, b) that rapamycin increases both ACD and memory generation in total or naïve CD8 T cells from aged mice, c) that virtual memory CD8 T cells (T_{vm}) display ACD that is not affected by rapamycin or AKTi. In general, the data look robust, although there are a few issues and concerns. These are as follows.

1. Other studies have shown that the CD8^{low} population, following first division, is enriched in cells that become memory cells. Based on this, the authors wish to conclude that, by increasing ACD with rapamycin treatment, they increase the numbers of these CD8^{low} cells and therefore memory. There is, however, another possibility: CD8^{high} cells produced in the first division do not normally contribute to memory, but do so when treated with rapamycin (as shown by Jonathan Powell and others). Unlike most other studies of ACD in CD8 T cells, the authors do not examine the relative contributions of the CD8^{hi} versus CD8^{lo} first division T cells. Without this, it is difficult to resolve these possibilities. Thus, it will be important to do these sorts with or without rapamycin treatment,

and assess contribution to memory, both for naïve and T_{VM} cells. Without such experiments, the conclusion, as stated in the title, is only supported by correlation.

The Powell group indeed performed CD8^{lo} and CD8^{hi} transfer experiments reported in Pollizzi et al., 2016, Nature Immunology. However, these cells were not previously treated with rapamycin. In the referred publication, the authors actually showed that mTOR activity is asymmetric upon mitosis, that mTOR is inherited by the CD8^{hi} cell (being the reason why they used CD8 expression as a readout for the adoptive transfers) and that translocation of mTOR to the lysosomes is dependent on amino acid uptake, while it remains unaffected by rapamycin treatment.

A contemporary manuscript, from the lab of Douglas Green (Verbist et al., 2016, Nature), where asymmetric metabolism was also addressed, indeed showed that *in vivo* inhibition of mTORC1 in recipient animals with rapamycin restored the ability of CD8^{hi}/cMyc^{hi} T cells to contribute to a secondary response. However, the strategy used by us, where mTOR inhibition was transient and done prior to adoptive transfer, certainly does not reflect in better contribution of CD8^{hi} cells to the memory pool. For naïve T cells, we have previously published data that would follow the reviewers' suggestion (Borsa et al., 2019, Science Immunology). There we could show that CD8^{hi} cells, generated by enforced ACD conditions (mitosis following transient rapamycin treatment), do not show memory potential to the same extent as CD8^{lo} cells. In fact, they behaved very similarly to CD8^{hi} cells emerging from untreated mother cells.

In addition, in T_{VM}'s there is no difference in ACD rates in presence or absence of rapamycin. Furthermore, in Suppl. Fig. 3, we show that in general T_{VM} cells have high mTOR activity, even in presence of inhibitors. Therefore, as already discussed with the editor, we believe that the proposed experiment would not add additional insights.

2. While numbers of P14 T cells are enumerated in recipient, challenged mice in Figures 2 and 5, only percentages are provided for subsets and cytokine production, necessitating recalculation by the reader (which can only be estimated, and significance is not addressable). Thus, when % T_e, T_{em}, and T_{cm} (or TNF, or IFN γ producing cells) show no difference (or some difference), what we really need to know is how many of these there are, and if the differences are significant. The same will apply to the experiments requested in #1.

We added numbers for subsets and cytokine-producing cells as requested in Figures 2 and 5.

3. The values for % TNF producing cells in Figure 2 and Figure 5 are strikingly different. Such experimental variation is troubling for any conclusions.

We have observed differences in the frequencies of cytokine producing cells among experiments. This might be due to fitness of the cells and variations in the staining. However, we do not believe this is a critical point. As we discussed previously, we do not want to suggest higher ACD rates or rapamycin treatment affect per cell cytokine production. Copying the answer to question number 7 from Reviewer 1: *"We previously reported that transient mTOR inhibition (and enforced ACD) overall does not change the ability of T cells to produce cytokines, but increases the frequencies of cells that survive for a long period in absence of antigenic stimulation (memory-like daughter cells) (Borsa et al., 2019, Science Immunology). This leads to differences in numbers (of cytokine producing cells)..."*

However, accounting for the reviewer's concern, we re-analysed our raw data and changed the gating for the TNF⁺ cells in the mentioned experiment (Fig.2) to the one below (Fig. F) (less stringent than previously used):

New gating

Figure F. New gating strategy for TNF producing cells.

The frequencies of TNF producing cells would then look more similar to the ones seen in Fig. 5, as depicted below (Fig. G, left panel). Furthermore, the frequencies would also be compatible to the ones seen in the same experiment repetition. Pooled data from experiment 1 and 2 concerning TNF expression can also be seen below (Fig. G, right panel). We corrected the figure accordingly.

Figure G. Corrected frequencies of TNF producing cells concerning animals analysed in Fig.2 of the manuscript (left panel) and pooled data from 2 experiments (right panel, when no statistical significance is observed, $p=0.0595$).

4. Often, experiments are only done twice, and in most cases do we know how many individual animals were used. For example, in Figure 3B, one cell division of two is shown. We certainly need to know how general the effects are.

We addressed this concern above (comment 9, reviewer 1).

5. The results in Figure 3 are interesting (and the Seahorse experiments are excellent), but not really integrated into the rest of the paper (other than to cite results from others). How does rapamycin treatment affect these differences? Are the effects related to ACD? (That is, are the CD8^{hi} or CD8^{lo} cells responsible for these differences?)

The seahorse experiments were performed aiming to better understand the physiology of T_{VM} progenies (which divide asymmetrically). As rapamycin treatment did not affect ACD rates in T_{VM} cells, we did not assess rapamycin treatment effects in the Seahorse experiments. However, it is clear from the comments of all the reviewers that we need to better discuss the metabolic profile of the T_{VM} compartment in the text, which has been done in the revised version of the manuscript.

While not aiming to link the metabolic profile of activated T_{VM} cells to any pharmacological modulation, we have now better discussed the metabolic phenotype of these cells. Quinn and collaborators (Quinn et al., 2020, Nature Communications) have recently described that T_{VM} cells can exhibit high spare respiratory capacity (SRC), which we also observed in activated T_{VM}'s, and that this correlates with IL-15 sensitivity and Bcl-2 expression, which can be associated with their superior survival capacity. Additionally, we can speculate about a possible link between higher ACD rates and the mixed metabolic profile observed for T_{VM} progenies. Strong ACD (as it occurs in T_{VM}'s) leads to polarized progenies of CD8^{hi} and CD8^{lo} daughter cells, which exhibit distinct metabolic profiles (Pollizzi et al., 2016, Nature Immunology). This might reflect – on a population level - in cells that are fully capable of performing both respiration and glycolysis, exactly what we observed and is depicted in Fig. 3.

6. A critical experiment is shown in Figure 5B, which is the only experiment that shows that numbers of old P14 cells are decreased upon challenge, and that this increases with rapamycin treatment to levels seen with young, rapamycin-treated cells. This experiment should be re-drawn to highlight this effect, which after all is the conclusion in the title of the paper. Further, as indicated in #1, the numbers rather than percentages for the subsets should be presented to highlight the effects. (see also #7).

We have added data about the numbers as requested.

7. A major basis for the studies herein is that T cells from aged mice generate less memory. They do not actually show us this point (in Figure 2 there is no comparison with young T cells). If, indeed, rapamycin increases memory to the level seen in young T cells (we do not know this—it is only shown for young, naïve T cells in Figure 5B), we may be able to attribute this effect to the increase seen in the naïve but not Tvm subset. For this, however, we need a direct comparison in which total, naïve, and Tvm are compared in an experiment such as that in Figure 5B.

As other reviewers pointed out, we did not show the comparison to young in Fig.2., but we believe this is not a critical issue, as in Fig.5 we perform a much more complete experiment separating cells in different subsets. We believe that the data shown in Figure 5, assessing the memory potential of the four subsets provides much better resolution than the suggested experiment using transfer of bulk populations.

Nevertheless, we discussed with the editor about the possibility of repeating this experiment. Because of the pandemic, we had to cull all aged P14 animals we had in the laboratory, as we could

not precisely estimate for how long the labs would be shut down back then – as we now also don't know for how long and how activities will be maintained. Thus, we would need to wait at least 18 months (with no further interruption in the activities) to be able to perform the suggested experiment. For now, we agreed with the editor that this experiment is not determinant for the article's main message. We would appreciate to have a similar understanding from the reviewers.

REVIEWER COMMENTS

Reviewer #1 (Remarks to the Author):

Many thanks to the authors for thoroughly addressing all comments, I commend the effort put into doing this, especially in these difficult times. I believe the authors have answered the majority of my concerns and have improved the paper substantially. It is a good read and the amendments made to the text make the story easy to follow.

I have two follow up points that might be good to address though overall I think this work should be published and I'm sure it will be well received.

Questions (numbered according to the rebuttal):

2) Regarding nomenclature of aged/young mice:

I accept the reasoning for this and it is sound logic. However, I do think it would aid the reader to more clearly state the ages of the mice used, or are the data derived from a mix of all of these mice? If so, perhaps exploring exact age in this context might be of interest given the skewed nature of the data in the old and old+R groups?

3) Regarding Fig. S1:

The data do not show this convincingly. Young+R t-bet is very variable and significance compared to young is not stated. T-bet asymmetry is shown in old+R vs old. However, P-S6 is significantly elevated in young+R compared to young, but not in old+R compared to old. There is a trend, but given the number of data points and the fact that this is mouse data I would expect this to be stronger. As this is supposed to be a control and supported by previous observations, it is not entirely convincing.

Reviewer #2 (Remarks to the Author):

The authors have done a very good job of addressing my queries and those of the other reviewers. The data in Figure S5 is convincing and helpful to validate the application of their observations in a non-TCR transgenic context. The addition of Figure S6 and increased discussion of survival is helpful for the field. I would still recommend specifically noting their P14 mice were Rag sufficient in the methods section- just to make sure interpretation is clear for the reader.

Reviewer #3 (Remarks to the Author):

The authors have tried to address my concerns, mostly by argument. Assuming that this is in agreement with the editors, I will not override the editor's decision.

A major conclusion of the paper, that memory is impaired in aged mice because ACD is impaired, remains unsupported by any data in the paper. A second major conclusion, that impairment of ACD impairs memory formation, is also unsupported. I do not think that the authors should state these as conclusions, but rather discuss them as untested possibilities.

REVIEWERS' COMMENTS

Author's responses in blue

Reviewer #1 (Remarks to the Author):

Many thanks to the authors for thoroughly addressing all comments, I commend the effort put into doing this, especially in these difficult times. I believe the authors have answered the majority of my concerns and have improved the paper substantially. It is a good read and the amendments made to the text make the story easy to follow.

We would like to thank the reviewer for the positive feedback.

I have two follow up points that might be good to address though overall I think this work should be published and I'm sure it will be well received.

Questions (numbered according to the rebuttal):

2) Regarding nomenclature of aged/young mice:

I accept the reasoning for this and it is sound logic. However, I do think it would aid the reader to more clearly state the ages of the mice used, or are the data derived from a mix of all of these mice? If so, perhaps exploring exact age in this context might be of interest given the skewed nature of the data in the old and old+R groups?

We agree that age might have an impact on data variability. Following the reviewer's suggestions, we have added the age of the mice used in each experiment in the Figure legends.

3) Regarding Fig. S1:

The data do not show this convincingly. Young+R t-bet is very variable and significance compared to young is not stated. T-bet asymmetry is shown in old+R vs old. However, P-S6 is significantly elevated in young+R compared to young, but not in old+R compared to old. There is a trend, but given the number of data points and the fact that this is mouse data I would expect this to be stronger. As this is supposed to be a control and supported by previous observations, it is not entirely convincing.

We agree with the reviewer about the high variability presented by the young and young+R groups concerning T-bet asymmetric inheritance. However, we believe that with a higher number of imaged mitoses the observed trend (more asymmetry upon rapamycin treatment) would reach statistical significance, as in our previous publication (Borsa et al., 2019, Science Immunology) we showed that transient mTOR inhibition in naïve CD8 T cells from young mice leads to increased CD8 and T-bet asymmetry. Our aim in the present work focused on evaluating whether rapamycin treatment would also be beneficial for ACD in cells from aged animals. Thus, we would like to highlight that the higher asymmetry of T-bet upon rapamycin treatment in T cells from young mice is not supported just by the data depicted in S1. Concerning the asymmetry of P-S6, we agree with the comment made by the reviewer. We have corrected the text accordingly aiming to avoid any strong statement about rapamycin modulation of asymmetric inheritance of both T-bet and P-S6.

Reviewer #2 (Remarks to the Author):

The authors have done a very good job of addressing my queries and those of the other reviewers. The data in Figure S5 is convincing and helpful to validate the application of their observations in a non-TCR transgenic context. The addition of Figure S6 and increased discussion of survival is helpful

for the field. I would still recommend specifically noting their P14 mice were Rag sufficient in the methods section- just to make sure interpretation is clear for the reader.

We would like to thank the reviewer for the positive feedback and once more for the very constructive insights made in the first round of revisions. We have added the information about P14 mice being Rag sufficient in the Methods section.

Reviewer #3 (Remarks to the Author):

The authors have tried to address my concerns, mostly by argument. Assuming that this is in agreement with the editors, I will not override the editor's decision.

We would like to thank the reviewer for all the suggestions done in the first round of revisions. As noted by the reviewer, we were not able to perform some of the experiments as a consequence of the pandemic, and we are very grateful for his/her understanding.

A major conclusion of the paper, that memory is impaired in aged mice because ACD is impaired, remains unsupported by any data in the paper. A second major conclusion, that impairment of ACD impairs memory formation, is also unsupported. I do not think that the authors should state these as conclusions, but rather discuss them as untested possibilities.

We agree that our conclusions are supported mainly by correlation. This was also the case on our previous publication about how improving ACD has a positive impact on memory formation (Borsa et al., 2019, Science Immunology). There we could show that by prohibiting cells to undergo ACD (which is achieved by blocking polarisome formation), transient rapamycin had no impact on both CD8 asymmetry and on memory formation. Besides being an additional correlation, it suggests that the effect of rapamycin on ACD is a contributing factor to memory formation. Nevertheless, we tried to be very careful to avoid any misleading overstatements and have modified any sentences where ACD modulation by rapamycin could be interpreted as having a direct causal effect on memory formation.